# Post-processing methods for delay embedding and feature scaling of reservoir computers
Jonnel Jaurigue [1] ✉, Joshua Robertson [2], Antonio Hurtado [2], Lina Jaurigue[1] & Kathy Lüdge [1]

Reservoir computing is a machine learning method that is well-suited for complex time series prediction tasks. Both delay embedding and the projection of input data into a higher-dimensional space play important roles in enabling accurate predictions. We establish simple post-processing methods that train on past node states at uniformly or randomly-delayed timeshifts. These methods improve reservoir computer prediction performance through increased feature dimension and/or better delay embedding. Here we introduce the multi-random-timeshifting method that randomly recalls previous states of reservoir nodes. The use of multi-random-timeshifting allows for smaller reservoirs while maintaining large feature dimensions, is computationally cheap to optimise, and is our preferred post-processing method. For experimentalists, all our post-processing methods can be translated to readout data sampled from physical reservoirs, which we demonstrate using readout data from an experimentally-realised laser reservoir system.

Reservoir computing is a machine learning method that exploits the dynamics of a reservoir for nonlinear mapping of inputs into a higher-dimensional space[1,2]. This exploitation of reservoir dynamics means that only the weights coupled to the final prediction output need to be trained. This elementary training approach, combined with the fact that a reservoir can be any dynamical system with finite memory showing consistent input responses, means that reservoir computers are well-suited for hardware implementation. In this paper, we will apply simple post-processing methods to the elementary training scheme. We will then present a post-processing method that improves reservoir computer performance for minimal computational investment. Finally, we translate our post-processing methods to the training scheme of a physical reservoir computer. This reservoir is implemented in photonic hardware using high-speed light signals for information processing.

Our post-processing methods are inspired by delay embedding. This is the process in which delayed replicas of a time series can be used to reconstruct the full state-space dynamics of the underlying dynamical system[3]. In a reservoir computer, the sampled driving input is nonlinearly mapped, and the resultant higher-dimensional representation is observed as a readout of reservoir states evolving in response to the driving input. Our simple post-processing methods train on delayed versions of these reservoir states, analogous to delay embedding, in order to better capture the driving system and thus improve predictive performance. Our methods can apply the delayed states in a uniform-timeshifting[4,5] or random-timeshifting[6,7]

manner, and will do so only through state rearrangement or replication. This notably preserves the fundamental benefit of using reservoir computers by keeping nonlinear mapping restricted only to the reservoir[8].

Uniform-timeshifting of reservoir states was investigated by Marquez et al.[4], who concatenated each readout state vector with a delayed replica of itself. They drew a connection to Taken's delay embedding[9] and showed that optimal delays are required. Picco et al. recently demonstrated that such uniformly timeshifted delays need not be evenly spaced[10]. The work of Del Frate et al.[6] introduced random-timeshifting. This approach stores the readout of reservoir states, and the feature vector used for training is comprised of states randomly sampled between the current time and some appropriately chosen maximal delay. Carroll and Hart[7] subsequently connected the performance improvement due to random-timeshifting with an increased covariance rank of the training state matrix.

Reservoir dynamics might also implicitly perform a delay embedding of a time series input. Reservoir memory is particularly important for optimal delay embedding and is dependent on the driving input's measured precision, input timescale and the prediction task. In such cases, issues relating to reservoir memory can be difficult to compensate for via hyper-parameter optimisation[11,12]. For example, for a finely discretised input, the optimal delay embedding may be larger than the timescale of the memory of the reservoir decays. Moreover, given that the most commonly used reservoirs are networks of randomly connected nodes, such architectures only allow for indirect tuning of reservoir memory through

[1]Institut für Physik, Technische Universität Ilmenau, Ilmenau, Germany. [2]Institute of Photonics, SUPA Department of Physics, University of Strathclyde, Glasgow, UK. ✉e-mail: jonnel-anthony.jaurigue@tu-ilmenau.de

hyperparameters such as the spectral radius. Conversely, time-multiplexed delay-based reservoirs allow for direct tuning through the reservoir's internal delay. Indeed, we previously demonstrated predictive performance improvement by optimising the internal delay of a time-multiplexed reservoir[5]. Similar trends are shown in other works[13–15]. However, in this paper, we will focus on post-processing methods that are applied after reservoir states have already been generated, circumventing difficulties associated with reservoir hyperparameter optimisation. Our post-processing focus also contrasts our previous work that applied pre-processing methods at the level of the driving input. We demonstrated that adding an optimally-delayed replica of the driving input as it is fed into the reservoir greatly improved performance. That approach could be considered a hybrid of reservoir computing and nonlinear vector regression, if the additional delayed input is viewed as training feature selection[16].

In the following sections, we will establish some simple post-processing methods on our simulated reservoir computer, for a variety of chaotic attractor prediction tasks. These are the aforementioned uniform-timeshifting and random-timeshifting methods. We also expand into multi-uniform-timeshifting that takes multiple uniformly-delayed states for training. Based on key findings regarding predictive performance when we recall past states using these simple post-processing methods, we introduce our multi-random-timeshifting method. This method scales well with larger feature dimensions, is computationally cheap to implement and can outperform all other post-processing methods.

We will then translate our established methods to an experimentally-realised reservoir computing system built with photonic hardware[17]. Such physical reservoir computers utilise complex nonlinear dynamics of physical systems for high-speed and energy-efficient analogue computing. Examples of physically implemented reservoirs include optical, optoelectronic and micro-mechanical systems[18–21], among others. Although good computing performance has been demonstrated for a number of tasks[10,22–25], this is often reliant on computationally expensive hyperparameter optimisation. There have been numerous studies into efficient methods of hyperparameter optimisation[26–30], however, there are often drawbacks which include the computation cost, lack of applicability to hardware reservoirs and a lack of generalisability to a wide range of tasks. Fortunately, our post-processing methods are applied after reservoir states have already been generated. We show that our methods are able to improve the predictive performance of physical reservoir computers, and circumvent the difficulties associated with physical reservoir hyperparameter optimisation.

## Methods
### Reservoir computer model
Figure 1 illustrates the general architecture of our simulated reservoir computer. We feed in a 1-dimensional input-series $x$ of $K$ steps into the reservoir, and our output is a prediction of the 1-dimensional target series $y$ of $K$ steps. For every input-step $x_i \in x \in \mathbb{R}^K$ driving the reservoir there are a total of $n$ nodes through which we observe the corresponding reservoir states vector $s_i = (s_{i,1}, ..., s_{i,n}) \in \mathbb{R}^n$. For the entire input-series $x$ of $K$ input-steps $(x_1, ..., x_K)$ we generate a total of $K$ sampled reservoir states vectors $(s_1, ..., s_K)$.

The focal methods of this paper relate to training and post-processing methods on the state matrix $S$. To generate the state matrix $S$, the sampled reservoir states $(s_1, ..., s_K)$ are collected into a state matrix

$$S = \begin{bmatrix} s_1 \\ s_2 \\ ... \\ s_{K-1} \\ s_K \end{bmatrix} = \begin{bmatrix} s_{1,1} & ... & s_{1,n} & 1 \\ s_{2,1} & ... & s_{2,n} & 1 \\ ... & ... & ... & 1 \\ s_{K-1,1} & ... & s_{K-1,n} & 1 \\ s_{K,1} & ... & s_{K,n} & 1 \end{bmatrix} \quad (1)$$

where the $i$-th row is the vector $s_i$ of $n$ states $(s_{i,1}, ..., s_{i,n})$ driven by input-step $x_i$. A bias term of one is appended to each row for training. The predicted

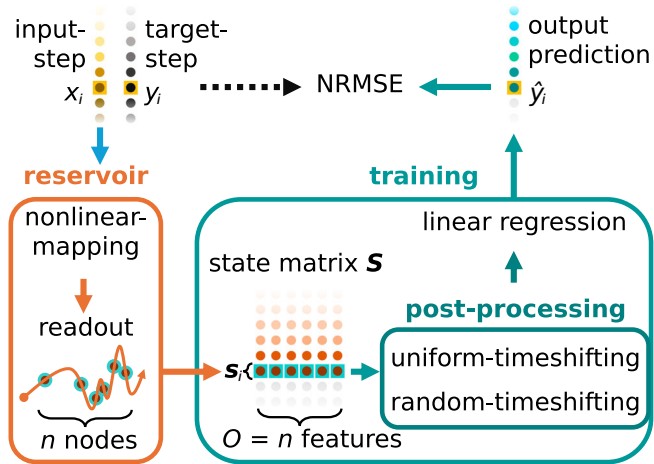

**Fig. 1 | General architecture of our simulated reservoir computer.** Input steps $x_i \in x \in \mathbb{R}^K$ are mapped through the reservoir of $n$ nodes. $O = n$ features of state matrix $S$ are used for training. Post-processing may increase the number of features $O$. The prediction output $\hat{y}_i$ is measured to target $y_i$ using normalised root mean square error (NRMSE).

target-series output of the reservoir $\hat{y}$ is then given by

$$S w = \hat{y}$$

where $w$ is the vector of $n + 1$ output weights. The weights $w$ are found by minimising the difference[31] between prediction $\hat{y}$ and the target $y$. The solution to this minimisation problem is analytically solved using ordinary least squares in matrix form

$$w = (S^T S + \lambda I)^{-1} S^T y$$

on a training set of $k < K$ sequential rows of the state matrix $S$.

To evaluate the predictive performance of our trained model, we use the normalised root mean square error (NRMSE) to quantify how accurate the predicted target-series $\hat{y}$ is to target-series $y$ for both the training and testing sets. NRMSE is given by

$$NRMSE = \sqrt{\frac{\sum_{i=1}^{k} (y_i - \hat{y}_i)^2}{k \hat{\sigma}^2}}$$

where $k$ is the row dimension of the training or testing set of the state matrix $S$, and $\hat{\sigma}^2$ is derived from the set's target $y$ as an approximation of the true population variance. For simulations that undergo repeat realisations, the median NRMSE ± median absolute deviation (MAD) is recorded as the predictive performance for that realisation set.

The number of columns of state matrix $S$ used for training is the feature dimension $O$. For base training without further post-processing, feature dimension $O = n$, referring to the number of sampled states $(s_{i,1}, ..., s_{i,n}) \in s_i$ per input-step $x_i$ equal to the node dimension $n$. We will refer to this state matrix $S$ with $O = n$ features described in Eq. (1) as the base state matrix on which further post-processing methods will be applied.

### Post-processing the state matrix
This paper focuses on several post-processing methods applied to the base state matrix $S$ described by Eq. (1). These methods rearrange and/or replicate the base state matrix $S$ in order to generate a post-processed state matrix. Notably, our post-processing methods keep nonlinear input mapping restricted to the reservoir, thereby preserving the computational advantage of reservoir computing[8].

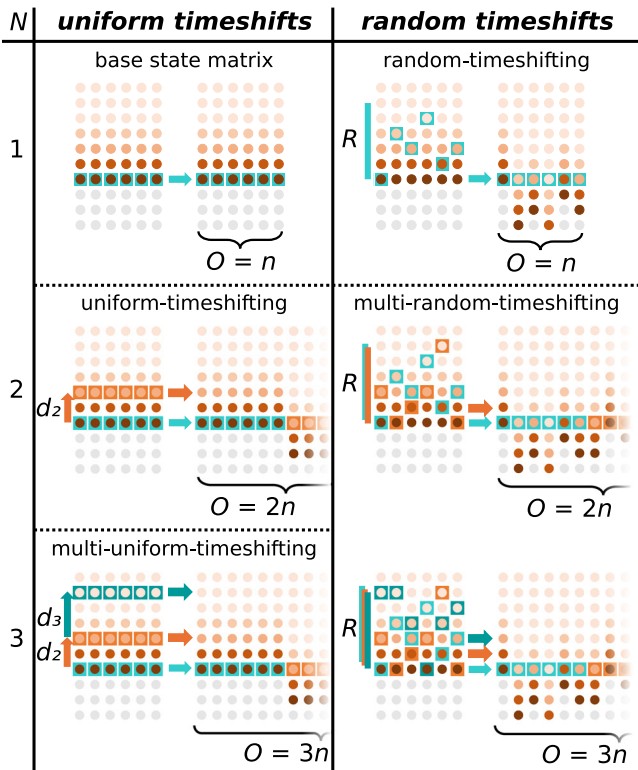

**Fig. 2 | Post-processing methods on the base state matrix S.** Each entry, reading left-to-right, indicates timeshifting of $S$ under a given post-processing method. $N = 1$ uniform-timeshifting indicates no post-processing.

**Uniform-timeshifting.** The uniform timeshifts post-processing method generates a state matrix that has a larger feature dimension $O$ compared to the base state matrix $S$. We start with a uniformly-timeshifted state matrix $S_{d_2}$, which is a replica of the base state matrix $S$ whose $O = n$ features are timeshifted $d_2$ input-steps into the past. This uniformly-timeshifted state matrix $S_{d_2}$ is then horizontally concatenated (denoted by $\frown$) to the base state matrix $S$, generating the uniform-timeshift state matrix

$$
S \frown S_{d_2} = \begin{bmatrix} s_1 \\ \dots \\ s_K \end{bmatrix} \frown \begin{bmatrix} s_{1-d_2} \\ \dots \\ s_{K-d_2} \end{bmatrix}
$$

$$
= \begin{bmatrix} s_{1,1} & \dots & s_{1,n} \\ \dots & \dots & \dots \\ s_{K,1} & \dots & s_{K,n} \end{bmatrix} \frown \begin{bmatrix} s_{1-d_2,1} & \dots & s_{1-d_2,n} \\ \dots & \dots & \dots \\ s_{K-d_2,1} & \dots & s_{K-d_2,n} \end{bmatrix}.
$$

In effect, any feature vector $s_i \in S$ is extended with the feature vector $s_{i-d_2} \in S$, where $d_2$ is the uniform time shift applied to the feature vector, illustrated by Fig. 2 at $N = 2$ state matrix replicas. In this way, the uniform-timeshift state matrix $S \frown S_{d_2}$ that is used for training and testing has an increased feature dimension of $O = 2n$. Earlier methods of concatenating uniformly timeshifted features have been previously established[4,5,32].

**Multi-uniform-timeshifting.** Inspired by delay embedding, we now take multiple timeshifts in order to better reconstruct the underlying dynamics of the driving system and improve predictive performance. The stepsize between multiple timeshifts can be of the same length, characteristic of delay embedding[3], or of varying length as demonstrated by Picco et al. under their "Timesteps Of Interest" algorithm[10].

Horizontally concatenating multiple uniformly-timeshifted state matrices generates a multi-uniform-timeshifts state matrix

$$
S \frown S_{d_2} \frown S_{d_3^{\Sigma}} = \begin{bmatrix} s_1 \\ \dots \\ s_K \end{bmatrix} \frown \begin{bmatrix} s_{1-d_2} \\ \dots \\ s_{K-d_2} \end{bmatrix} \frown \begin{bmatrix} s_{1-d_2-d_3} \\ \dots \\ s_{K-d_2-d_3} \end{bmatrix}
$$

of $O = 3n$ features, where $d_3^{\Sigma}$ indicates the sum of iterative timeshifts $d_2 + d_3$ (Fig. 2 at $N = 3$ replicas). For example, if uniform-timeshifts $d_2 = 2$ and $d_3 = 8$, then the first replica is the base state matrix $S$, the second replica is the uniformly-timeshifted state matrix $S_{d_2}$ at $d_2 = 2$ input-steps into the past and the third replica is the uniformly-timeshifted state matrix $S_{d_3^{\Sigma}}$ at $d_2 + d_3 = 2 + 8 = 10$ input-steps into the past.

Concatenating $N$ state matrix replicas generates a multi-uniform-timeshifts state matrix $S \frown S_{d_2} \frown S_{d_3^{\Sigma}} \frown \dots \frown S_{d_N^{\Sigma}}$ of $O = Nn$ features, with the earliest state matrix replica $S_{d_N^{\Sigma}}$ timeshifted $d_2 + d_3 + \dots + d_N$ input-steps into the past. Allowing for an increased number of delayed states should in general enable an improved embedding. The increased computational cost associated with the increasing number of optimisation hyperparameters will be discussed.

**Random-timeshifting.** The random-timeshifting post-processing method is implemented according to published methods[6]. Here, each of the $O = n$ features of the base state matrix $S$ is timeshifted a number of input steps into the past. The timeshift for each feature is randomly chosen. This random-timeshifts state matrix $S_{r^1}$ can be described as

$$
S_{r^1} = \begin{bmatrix} s_{1-r^1} \\ \dots \\ s_{K-r^1} \end{bmatrix} = \begin{bmatrix} s_{1-r_1^1,1} & \dots & s_{1-r_n^1,n} \\ \dots & \dots & \dots \\ s_{K-r_1^1,1} & \dots & s_{K-r_n^1,n} \end{bmatrix}
$$

where the random-timeshifts vector $r^1$ consists of random integers $r_1^1, \dots, r_n^1$ ranging from 0 to some max-random-timeshift $R$. Thus, the $j$-th feature of the base state matrix $S$ is timeshifted $r_j^1 \in r^1$ input-steps into the past, illustrated by Fig. 2 at $N = 1$ state matrix replica. The feature dimension of this random-timeshifts state matrix $S_{r^1}$ remains at $O = n$.

The max-random-timeshift $R$ parameter sets the range of random-timeshifts $r \in r$ that may be randomly chosen. It is analogous to the method used by Del Frate et al., where the half-life of the autocorrelation function is multiplied by a real scalar[6] to determine the range of allowed random timeshifts $r \in r$. The max-random-timeshift $R$ can be optimised with a one-dimensional scan.

**Prediction tasks**

Various chaotic time series prediction tasks are performed on the simulated reservoir to establish and quantify our post-processing methods in this manuscript. Our driving input series is a coordinate of one of the Lorenz, Rössler, or Mackey–Glass attractors. Our target series is another coordinate of the attractor for cross-prediction tasks, or the input series coordinates a number of steps ahead for forecasting tasks. For configuring our simulated reservoir computer and establishing our post-processing methods we carried out prediction tasks on the Lorenz attractor.

Lorenz attractor driving input-series $x = [x_1, \dots, x_K]$ was the Lorenz $x$-coordinate defined by

$$
\frac{dx}{dt} = a(y - x),
$$
$$
\frac{dy}{dt} = x(b - z) - y,
$$
$$
\frac{dz}{dt} = xy - cz,
$$

and cross-prediction target-series $y$ was the Lorenz $z$-coordinate[33].

**Table 1 | Task and reservoir parameters**

| Lorenz attractor task parameters | | |
|---|---|---|
| $a = 10$ | $b = 28$ | $c = \frac{8}{3}$ |
| Integration timestep d$t = 0.001$ | | |
| Signal discretision timestep = 0.02 | | |
| Mackey–Glass attractor task parameters | | |
| $n = 10$ | $\beta_0 = 0.2$ | $\tau_{MG} = 17$ |
| $\theta = 1$ | $\gamma = 0.1$ | |
| Integration timestep d$t = 0.01$ | | |
| Signal discretisation timestep = 1 | | |
| Simulated reservoir parameters | | |
| $\theta = 1$ | $a = 40$ | $b = 0.025$ |
| $g = 0.05$ | $m \sim U [0, 1)$ | $\tau = n + \theta$ |
| $n = 60$, unless indicated otherwise | | |
| Realisations = 30 | | |
| Input $x_i mg$ scaled to the interval [0,1] | | |
| Total $K = 35{,}000$ | Training = 10,000 | Testing = 5,000 |
| Pre-training buffer = 10,000 | | |
| Pre-testing buffer = 10,000 | | |
| Uniform-timeshifts regularisation $\lambda = 10^{-8}$ | | |
| Random-timeshifts regularisation $\lambda = 10^{-7}$ | | |
| VCSEL physical photonic reservoir parameters | | |
| $n = 325$ | | |
| Realisations = 30 | | |
| Total $K = 10{,}000$ | Training = 6,000 | Testing = 3,000 |
| Pre-training buffer = 500 | | |
| Pre-testing buffer = 500 | | |
| Uniform-timeshifts regularisation $\lambda = 0.0$ | | |
| Random-timeshifts regularisation $\lambda = 1.0$ | | |

The simulated reservoir training models established on the Lorenz attractor were corroborated with Mackey–Glass attractor $P$-to-$P$ 10-step-ahead forecasting tasks (Suppl. Note 3). Mackey–Glass attractor driving input-series $x$ was the Mackey–Glass $P$-coordinate defined by

$$\frac{dP}{dt} = \frac{\beta_0 \theta^n P(t - \tau_{MG})}{\theta^n + P(t - \tau_{MG})^n} - \gamma P(t)$$

and forecasting target-series $y$ was the 10-step-ahead Mackey–Glass $P$-coordinate[34]. All time series coordinates were iterated using 4th-order Runga–Kutta. For the delay terms in the Mackey–Glass equation cubic Hermitian interpolation was used. The system parameters for the Lorenz and Mackey–Glass attractors are given in Table 1. Cross-prediction tasks on the Rössler attractor were also carried out in order to corroborate our established methods (Suppl. Note 4).

The simulated reservoir training models were translated to experimentally-realised readout data acquired from a laser-based photonic reservoir system. Targets were $P$-to-$P$ 1-step-ahead or 10-step-ahead coordinates of the input series, corresponding to a Mackey–Glass attractor.

## Reservoir systems
**Simulated reservoir.** For our simulated reservoir, we use a time-multiplexed nonlinear delayed map introduced by Jaurigue et al.[5,16]. This system describes a semiconductor optical amplifier subject to weak self-

feedback given by

$$s_{t+\theta} = \frac{a(bs_{t-\tau} + x_i mg)}{1 + bs_{t-\tau} + x_i mg}$$

with simulated reservoir parameters specified in Table 1. The input-series $x$ is masked ($m$) and scaled ($g$). The evolving reservoir state over time $s_t$ is observed once during each mask interval $\theta$ and then indexed into the state vector ($s_{i,1}, \ldots, s_{i,n}$) driven by input-step $x_i \in x$. There are a total of $n$ nodes through which we observe the reservoir state $s \in s_i$, distinguished by their unique mask $\theta$, so the length of a clock cycle driven by input-step $x_i$ is $n\theta$.

Given that we apply a random mask on the driving input for our simulated reservoir[5], each reservoir computer configuration undergoes repeat realisations, with each realisation using a different random mask. When applicable, each realisation also uses a different vector of random timeshifts $r$. For specific details on our simulated reservoir and the time-multiplexed masking procedure, we refer the reader to previous work[5], given that the reservoir has no influence on the post-processing methods which are the focus of this paper.

**Physical photonic reservoir.** To investigate post-processing methods on a physical reservoir system we utilised the readout of our previously published photonic reservoir based on a vertical-cavity surface-emitting laser (VCSEL) system[17]. This system enabled operation with high-speed (GHz-rate) and low-power (sub-mW) optical input signals. This system also enabled a hardware-friendly approach using a single VCSEL requiring very low bias currents, operating at a key optical telecom wavelength. The referred VCSEL-based photonic reservoir demonstrated good performance in data classification and time series prediction tasks, across a wide parameter space range. Physical reservoir parameters are in Table 1. For specific details on the physical reservoir, we refer the reader to our relevant work[17].

We used a single hardware-realised readout, rather than a set of realisations each using a different random mask, as is the case in our simulated reservoir computer. Multiple realisations were required in post-processing when using different vectors of random timeshifts $r$.

## Results
### Simple post-processing
We established the simulated reservoir computer configurations and post-processing methods by performing Lorenz $x$-to-$z$ cross-prediction tasks. Key results for characterising the performance of our post-processing methods are summarised in Table 2, which we reference in the following sections.

In Table 2, "N replicas" refers to the number of state matrix replicas used to generate the final state matrix used for training and testing. "$O$ features" refers to the feature dimension of this state matrix, expressed as multiples of the node dimension $Nn$. Results in the "uniform-timeshifting, $S^{\frown} \ldots ^{\frown} S_{d_N^{\Sigma}}$" category are for state matrices with uniformly-timeshifted features, which includes the base state matrix $S$ at $N = 1$. Results in the "random-timeshifting, $S_{r^1}^{\frown} \ldots ^{\frown} S_{r^N}$" category are for state matrices with randomly-timeshifted features. We present the median NRMSE $\pm$ MAD result under optimal delay configurations, for increasing replica number $N$. The reservoir is fixed at $n$ nodes.

**Uniform-timeshifting improved performance.** We start with results for uniformly timeshifted state matrices summarised in Table 2. For $N = 1$ replicas this corresponds to no post-processing on the base state matrix $S$. As expected, the base training method demonstrates inferior performance when compared to any of the optimised post-processing methods performed on the base state matrix $S$.

For $N = 2$ replicas, we horizontally concatenate a uniformly-timeshifted state matrix to the base state matrix in order to generate the uniform-timeshifted state matrix $S^{\frown}S_{d_2}$ of $O = Nn = 2n$ features. The optimal uniform-timeshift $d_2$ is also listed in Table 2 and was determined via

**Table 2 | Simulated reservoir computer results of Lorenz $x$-to-$z$ cross-prediction**

| $N$ replicas | $O$ features | Uniform-timeshifting, $S \frown \ldots \frown S_{d_N^\Sigma}$ | | | | | | Random-timeshifting, $S_{r^1} \frown \ldots \frown S_{r^N}$ | |
|---|---|---|---|---|---|---|---|---|---|
| | | NRMSE ± MAD | $d_2$ | $d_3$ | $d_4$ | $d_5$ | $d_6$ | NRMSE ± MAD | $R$ |
| 1 | $n$ | 0.065 ± 0.007* | – | – | – | – | – | 0.029 ± 0.005 | 19 |
| 2 | $2n$ | 0.017 ± 0.002 | 21 | – | – | – | – | 0.011 ± 0.002 | 32 |
| 3 | $3n$ | 0.0068 ± 0.0005 | 13 | 21 | – | – | – | 0.006 ± 0.001 | 40 |
| 4 | $4n$ | 0.0033 ± 0.0004 | 10 | 17 | 19 | – | – | 0.0040 ± 0.0005 | 44 |
| 5 | $5n$ | 0.0023 ± 0.0006 | 8 | 14 | 17 | 21 | – | 0.0028 ± 0.0007 | 57 |
| 6 | $6n$ | 0.0020 ± 0.0003 | 6 | 9 | 15 | 20 | 10 | 0.0024 ± 0.0004 | 55 |
| 10 | $10n$ | – | – | – | – | – | – | 0.0014 ± 0.0002 | 59 |
| 60 | $60n$ | – | – | – | – | – | – | 0.00055 ± 0.00003 | 94 |
| 120 | $120n$ | – | – | – | – | – | – | 0.00047 ± 0.00004 | 123 |

*NRMSE* normalised root mean square error, *MAD* median absolute deviation.

*Result of training on the base state matrix $S$.

a scan, illustrated in Fig. 3a (teal line). A 74% decrease in error was observed. Superior performance was observed even at sub-optimal configurations of uniform-timeshift $d_2$, which we attribute to the increase in the feature dimension $O$ from $n$ to $2n$. The superior performance of training on the optimally-delayed uniform-timeshift state matrix $S \frown S_{d_2}$, compared to training only on the base state matrix $S$ (orange dashed line), is consistent with previously published results[4,5].

In general, reservoir computer performance scales with the size of the internal and readout layers of its reservoir, provided there are sufficient readout nodes for adequately sampling the reservoir. This is reflected in the information processing capacity introduced by Dambre et al.[35], where it was shown that the maximal possible information processing capacity is equal to the readout node dimension $n$, which corresponds to the $O = n$ features of the base state matrix $S$. The improved performance of the uniform-timeshift state matrix $S \frown S_{d_2}$ can therefore, in part, be attributed to its increased feature dimension $O = 2n$, compared to the $O = n$ feature dimension of the base state matrix $S$. Viewed in this way, uniform-timeshifting is a computationally-inexpensive post-processing alternative to simulating larger reservoirs for higher-dimensional mapping of the input-series[36,37], while achieving similar results. In the case of physical reservoir systems, post-processing methods are sometimes the only viable option for increasing state matrix feature dimensions when reservoir parameters, such as node dimension, are constrained.

**Multi-uniform-timeshifting improved performance.** From the established uniform-timeshifting post-processing method, we proceed with multiple-uniform-timeshifting that increases the final state matrix feature dimension beyond $O = 2n$.

With $N = 3$ state matrix replicas we generate a uniform-timeshifts state matrix $S \frown S_{d_2} \frown S_{d_3^\Sigma}$ of feature dimension $O = 3n$. Table 2 lists the NRMSE for $N = 3$ replicas, referring to performance at optimal uniform-timeshifts $d_2$ and $d_3$ determined via a 2-dimensional scan. We observed a 60% decrease in error over $N = 2$ replicas. For $N$ replicas we generate uniform-timeshifts state matrices of feature dimension $O = Nn$, and the optimal uniform-timeshifts $[d_2, \ldots, d_N]$ are listed, determined via $(N - 1)$-dimensional scans. Going to $N = 4$, to $N = 5$ and to $N = 6$ replicas show a 51%, 30% and 13% decrease in error, respectively. Thus, a trend of increasingly improved predictive performance with increasing $O = Nn$ feature dimensions was observed.

In addition to increasing state matrix features $O = Nn$, performance with increasing $N$ replicas and optimal uniform-timeshifts $d$ is related to delay embedding[4,11,37]. The optimal values of the uniform-timeshifts $[d_2, \ldots, d_N] \in d$ are not the same, indicating that the optimal delay embedding from this task is when the uniform-timeshifts $d \in d$ are not evenly spaced. In general, not all delay embeddings work equally well when

the input data has only finite precision[3], as the theoretical guarantees of Taken's delay embedding theory (where delayed replicas of an input time series can be used to reconstruct the full state-space dynamics of the underlying dynamical system) only hold for infinite precision[9]. Furthermore, optimal delay embeddings depend on the criteria by which they are judged, which in our case is the NRMSE of the resulting cross-prediction. Moreover, since the reservoir itself performs a delay embedding[11,37,38], this affects the optimal delay embedding provided by multi-uniform-timeshifting of the reservoir states. Indeed, we observed that the optimal uniform-timeshift $d_2$ at $N = 2$ replicas changes with our simulated reservoir delay $\tau$ (Suppl. Note 3).

Finally, for multi-uniform-timeshifting, we show that the optimal configurations for the uniform-timeshifts $d$ are constrained to a single zone only. For example, Fig. 3b illustrates the results of a 2-dimensional scan for uniform-timeshifts $[d_2, d_3] \in d$ configurations when concatenating $N = 3$ replicas. We see superior performance for a single zone (dark teal/black area) of $[d_2, d_3] \in d$ configurations. Figure 3c also illustrates a single optimal configuration zone (dark teal/black area) for $N = 4$ replicas as was observed for $N = 3$. We do not observe multiple or repeating optimal zone patterns, illustrating that configurations must be specifically chosen for superior performance. This indicates the importance of thorough scans to determine the optimal uniform-timeshifts $d \in d$ configurations, which becomes computationally intractable with increasing $N$ replicas.

**Random-timeshifting improved performance.** Given our indications that multi-uniform-timeshifting requires optimally chosen uniform-timeshifts $d \in d$ configurations for superior performance, we looked to post-processing methods that perform well with randomly chosen timeshift parameters. Hence, we established the random-timeshifting post-processing method for our simulated reservoir computer. The "random-timeshifting, $S_{r^1} \frown \ldots \frown S_{r^N}$" column of Table 2 lists the result when training on the random-timeshifts state matrix $S_{r^1}$ at $N = 1$ replica. The optimal max-random-timeshift $R$ found via a parameter scan is listed alongside the NRMSE, with Fig. 3d illustrating this scan.

We report a 55% decrease in error when training on the optimal random-timeshifts state matrix $S_{r^1}$ ($N = 1$), compared to the base state matrix $S$ with the same feature dimension of $O = n$. This finding is consistent with the work of Carroll and Hart[7], and Del Frate et al.[6], who also demonstrated improved performance after random timeshifting on the base state matrix, measured by a reduced prediction error or a reduction in required reservoir state samplings per input.

The performance improvement as a result of random-timeshifting cannot be explained by a feature dimension increase as it is for uniform-timeshifting, because the random-timeshifts state matrix $S_{r^1}$ shares the same feature dimension of $O = n$ as the base state matrix $S$. Additionally, although

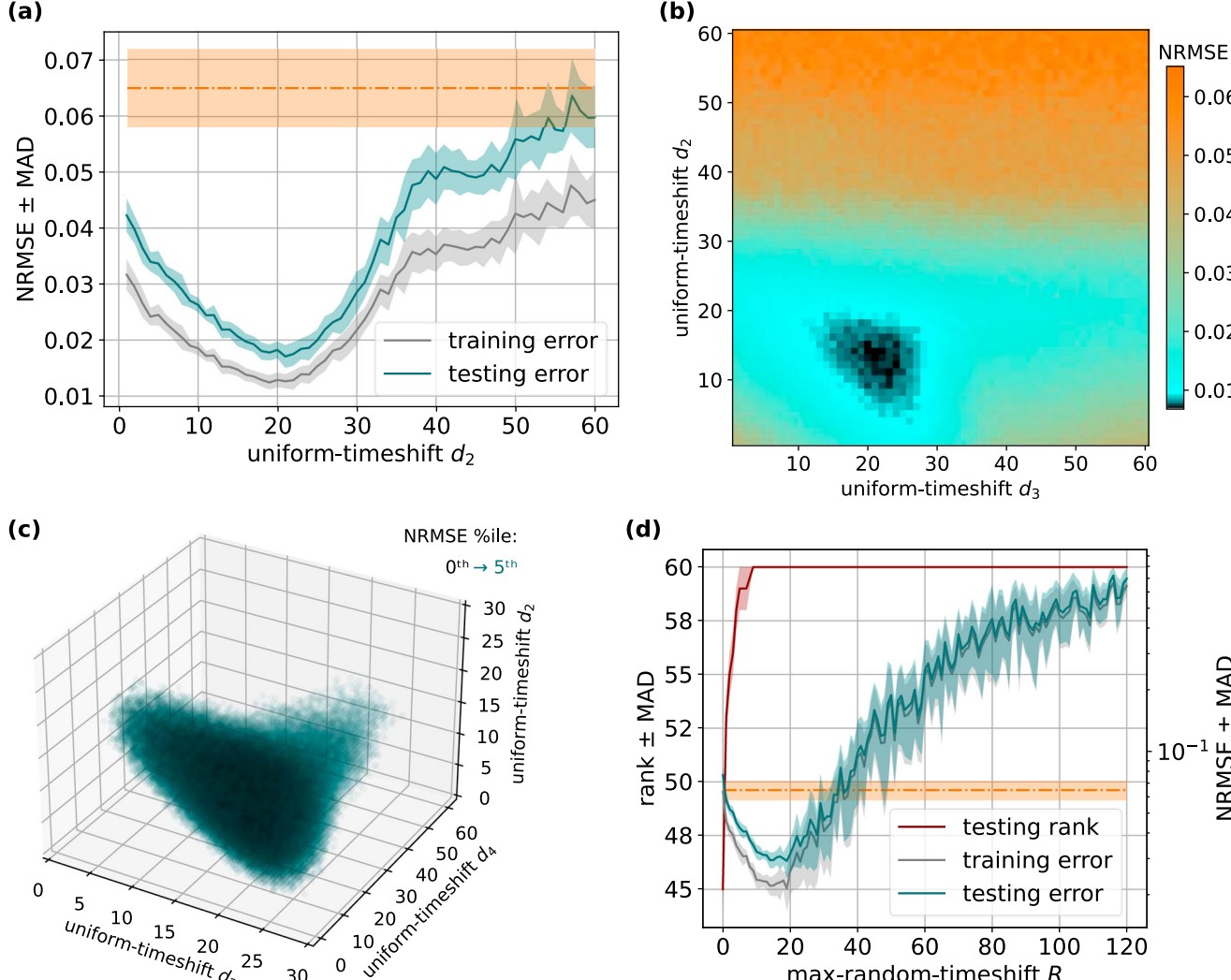

**Fig. 3 | Timeshifting on the simulated reservoir computer, Lorenz $x$-to-$z$ cross-prediction. a** Scan of uniform-timeshift $d_2$, for training error (grey line) and testing error (teal line) at $N = 2$ replicas and $O = 2n$ features. The orange dashed line indicates the result of training on the base state matrix $S$. The shaded area around each line indicates the MAD of the realisation set. **b** Scan of multi-uniform-timeshifts $d_2$ vs $d_3$, for testing error at $N = 3$ replicas and $O = 3n$ features. **c** Scan of multi-uniform-timeshifts $d_2$ vs $d_3$ vs $d_4$, for testing error at $N = 4$ replicas and $O = 4n$ features. **d** Scan of max-random-timeshift $R$ for covariance rank (maroon line), training error (grey line) and testing error (teal line) at $N = 1$ replica and $O = n$ features. The orange dashed line indicates the result of training on the base state matrix $S$. The shaded area around each line indicates the MAD of the realisation set. NRMSE normalised root mean square error, MAD median absolute deviation.

random-timeshifting introduces delayed versions of states, these timeshifted states replace the non-timeshifted base states as training features. Training on delayed replicas of states that append to the base state matrix, as we do in uniform-timeshifting, is more analogous to delay embedding.

Instead, a possible explanation for this performance improvement is that random timeshifting increases the covariance rank of the final training matrix, implying higher-dimensional mapping of the input[7,39]. Figure 3d shows how covariance rank (maroon line) and NRMSE (teal line) change with increasing max-random-timeshift $R$. Indeed, we see that covariance rank is larger at the optimal max-random-timeshift $R = 21$ compared to the base state matrix without random-timeshifting, consistent with previously published explanations[7,39]. However, the complete scan shows that the covariance rank of the random-timeshifts state matrix $S_{r^1}$ increases with max-random-timeshift $R$, and then the covariance rank remains high even as predictive performance degrades past the optimal max-random-timeshift $R > 21$. Note that changing the prediction task from $x$-to-$z$ cross-prediction to $x$-to-$x$ 1-step-ahead forecasting gives the same outcome, where optimal performance is achieved when covariance rank is minimal (Suppl. Note 2). Thus, a large covariance rank may not guarantee good performance, and the assumption that a state matrix with a larger covariance rank generally

outperforms a state matrix with a smaller covariance rank[7,12,39,40] warrants further investigation.

## Recall implementation

Here we will report our investigation of recall, which refers to delayed versions of states from the base state matrix $S$ being replicated in post-processing, in order to build up training state matrices of larger feature dimensions. For example, uniform-timeshifting that concatenates $N = 2$ state matrix replicas will result in states appearing up to $N = 2$ times in the uniform-timeshifts state matrix $S \frown S_{d_2}$. The recalled state replicas $S_{d_2}$ are delayed versions of those in the base state matrix $S$, separated by the uniform-timeshift $d_2$.

Reservoir computing for time series prediction seems to work by both delay embedding the input time series[11] and by projecting the input data into a higher-dimensional space[31]. The latter is generally related to the number of nodes of the reservoir. In the context of recall and uniform timeshifting, we show that reservoirs with a smaller node dimension perform better than reservoirs with a larger node dimension, which goes against the general assumptions that a higher-dimensional reservoir would result in better computational performance[35].

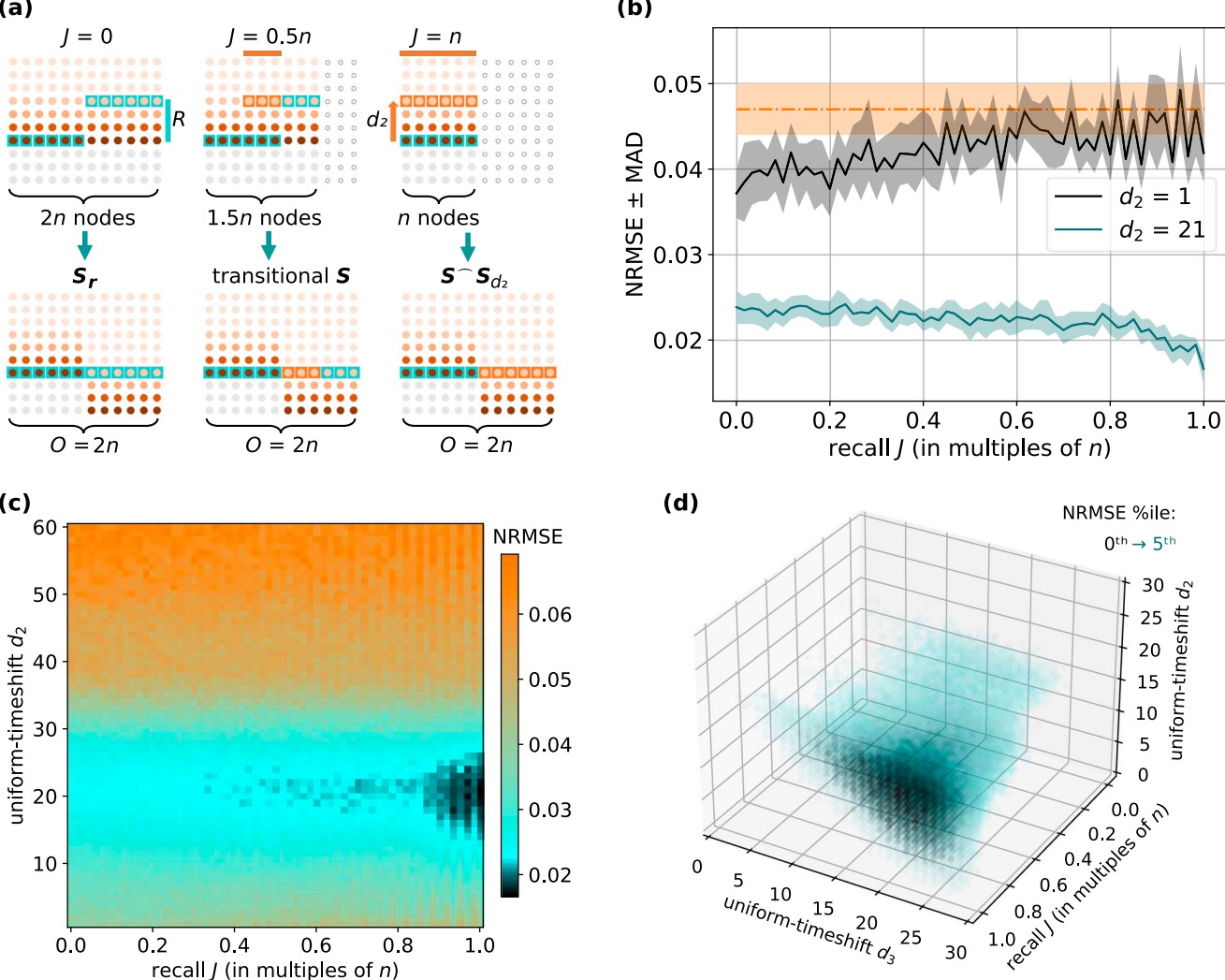

**Fig. 4 | Recall implementation on the simulated reservoir computer, Lorenz $x$-to-$z$ cross-prediction. a** Applying recall to the $2n$-extended base state matrix $S$. Null recall $J = 0$ generates a timeshift matrix $S_r$. Partial recall $J = 0.5n$ generates a transitional final state matrix. Total recall $J = n$ generates a uniform-timeshifts matrix $S^\frown S_{d_2}$. **b** Scan of recall $J$ vs uniform-timeshift $d_2 = 1$ (black line) and $d_2 = 21$ (teal line), for testing error at $O = 2n$ features. The orange dashed line indicates the result of training on the extended base state matrix $S$ of node dimension $2n$. The shaded area around each line indicates the MAD of the realisation set. **c** Scan of recall $J$ vs uniform-timeshift $d_2$, for testing error at $O = 2n$ features. **d** Scan of recall $J$ vs multi-uniform-timeshifts $d_2$ vs $d_3$, for testing error at $O = 3n$ features. NRMSE normalised root mean square error, MAD median absolute deviation.

To characterise the recall phenomenon, recall $J$ will indicate how many nodes will have sampled states recalled during post-processing, thereby introducing delayed versions of node states in the final state matrix. We start with an extended state matrix $S$ of feature dimension $O = 2n$, generated from a larger reservoir of $2n$ nodes. A possible configuration of timeshifts $[r_1, \ldots, r_{2n}] \in r$ applied to this extended state matrix $S$ is illustrated in Fig. 4a at recall $J = 0$. For this timeshifts vector $r$ the first $n$ timeshifts $[r_1, \ldots, r_n]$ are 0 and the second subset of $n$ timeshifts $[r_{n+1}, \ldots, r_{2n}]$ are at the max-timeshift $R$.

The constructed timeshifted state matrix $S_r$ in Fig. 4a is similar to the uniform-timeshift state matrix $S^\frown S_{d_2}$ at recall $J = n$. Here, both post-processed state matrices have $O = 2n$ features, and the uniform-timeshift $d_2$ is equal to max-timeshift $R$. The key difference between these state matrices is the number of nodes required to generate them. For the constructed timeshift state matrix $S_r$, the timeshifted features $[n + 1: 2n]$ are sampled states of the nodes $[n + 1: 2n]$, from a reservoir of $2n$ nodes. Conversely, for the uniform-timeshift state matrix $S^\frown S_{d_2}$, the timeshifted features $[n + 1: 2n]$ are generated by recalling the sampled states of nodes $[1: n]$, from a smaller reservoir of $n$ nodes. Specifically, the recalled node states are collected at $d_2 = R$ input steps into the past, thereby clamping the required node dimension to $n$. Thus, in Fig. 4a at null recall $J = 0$, we generate the

constructed timeshift state matrix $S_r$ from a readout layer of $2n$ nodes. Although the features $[n + 1: 2n]$ are timeshifted by $R = d_2$ input steps into the past, the final state matrix will have all states from the $2n$ nodes appearing only once. At total recall $J = n$, we generate the uniform-timeshift state matrix $S^\frown S_{d_2}$ from a readout layer of $n$ nodes. All $n$ nodes will have delayed replicas of their sampled states appearing in the final state matrix at $d_2 = R$ input steps into the past. A transitional state matrix at recall $J = 0.5n$ illustrates the case where only a third of the $1.5n$ nodes will have sampled states recalled.

**Total recall improves performance even as it reduces node dimension.** We have shown how simple post-processing methods can introduce delayed states into the training scheme that lead to predictive performance improvement at optimal timeshift configurations. For uniform-timeshifting and multi-uniform-timeshifting, we attributed this to both an improved delay embedding of the input and an increase in the number of training features $O$. Recall $J$ indicates how many nodes will have delayed replicas of sampled node states appearing in the final state matrix, and we investigated how recall affects prediction performance with delay embedding and feature dimension scaling effects in mind.

In Fig. 4b, we explored the effect of increasing recall $J$ at specific uniform-timeshift $d_2$ values. The left side of the graph indicates null recall $J = 0$, and the right side indicates total recall $J = n$ for generating the final state matrix of $O = 2n$ features, corresponding to the left-to-right orientation of the Fig. 4a schematics. We see that at sub-optimal timeshift $R = d_2 = 1$ (black line) performance suffers as recall $J$ increases. This is to be expected, since with increasing recall $J$ values from 0 to $n$ the node dimension of the reservoir decreases from $2n$ to $n$. However, the trend reverses unexpectedly at timeshift $R = d_2 = 21$ (teal line), where we reach peak performance at total recall $J = n$, corresponding to the smallest node dimension $n$.

To explain this phenomenon, we can look to delay embedding. The optimal delayed states introduced in post-processing are those that are timeshifted by $R = d_2 = 21$ input steps into the past. Thus, at null recall $J = 0$ we generate the constructed timeshifted state matrix $\boldsymbol{S_r}$ with max-timeshift $R = 21$, and observe improved performance over sub-optimal delays. As we increase from null recall $J = 0$ to total recall $J = n$, we increase the number of delayed state replicas used for the final training state matrix. By increasing this recall $J$, performance subsequently improves. Once we are at total recall $J = n$ we generate the uniform-timeshift state matrix $\boldsymbol{S} \frown \boldsymbol{S}_{d_2}$, which shows the best performance. Uniform-timeshift state matrix $\boldsymbol{S} \frown \boldsymbol{S}_{d_2}$ introduces delayed replicas of states at $d_2 = 21$ input-steps into the past, and in this way is a more analogous implementation of delay embedding. This is in contrast to the constructed timeshifted state matrix $\boldsymbol{S_r}$ which introduces delayed states from new nodes, rather than recalling states from the past. Viewed in this way, we consider recall $J$ as a way to scale the influence of a larger reservoir vs the influence of delay embedding. Therefore, the optimal amount of recall $J$ depends on whether the delay embedding is also optimal.

For completeness in Fig. 4b, we include the result of the state matrix with null recall $J = 0$ at $d_2 = 0$ (orange dashed line), which is equivalent to a base state matrix derived from a reservoir of node dimension $2n$. The inferior performance of the base state matrix derived from $2n$ nodes demonstrated that introducing even a sub-optimal uniform timeshift leads to better predictive performance.

In Fig. 4c, we see predictive performance when scanning increasing recall $J$ with uniform-timeshift $d_2$ values. We find superior performance (dark teal/black area) at total or near total recall $J \approx n$ when uniform-timeshift $d_2$ is optimal, even though the reservoir node dimension is at its lowest. Moreover, this superior performance at total or near total recall $J \approx n$ is demonstrated even when the readout node dimension is not rescaled, but kept at $2n$ (Suppl. Note 1), indicating that explicit delay embedding effects introduced in post-processing dominate any implicit changes in reservoir memory as a result of changing the node dimension from $2n$ to $n$.

Figure 4d illustrates the effect of recall with $N = 3$ replicas, generating the multi-uniform-timeshifts state matrix $\boldsymbol{S} \frown \boldsymbol{S}_{d_2} \frown \boldsymbol{S}_{d_3}$ of feature dimension $O = 3n$. Even in this higher-dimensional case, the optimal configurations of $d_2$ and $d_3$ demonstrate superior predictive performance (dark teal/black area) as we approach total recall $J = n$. The node dimension subsequently shrinks from $3n$ to $n$, indicating that increasing the influence of an optimal delay embedding overcomes any potential negative effect of a smaller reservoir.

Given that increasing recall $J$ reduces the reservoir node dimension $n$, we evaluated the effect of recall $J$ on the covariance rank of the state matrix, because it has been hypothesised that larger covariance ranks generally lead to superior predictive performance[7,12,39,40]. We observed that covariance rank derived from the same realisation set used for Fig. 4c is not correlated with zones of superior performance (Suppl. Note 1). This corroborates Fig. 3d, where the covariance rank of the random-timeshifts state matrix $\boldsymbol{S}_{r^1}$ monotonically increased with max-timeshift $R$.

## Multi-random-timeshifting

We now introduce multi-random-timeshifting. The motivation for this post-processing method is to increase the feature dimension of the final training state matrix, allow for better delay embeddings and utilise the lower optimisation cost of randomly-timeshifting features. We were inspired by recall, which scales the influence of delay embedding, and leads to improved performance as if we had increased reservoir size directly.

Multi-random-timeshifting recalls past states and increases the state matrix feature dimension beyond $O = n$, as is the case for uniform-timeshifting but not random-timeshifting. $N$ state matrix replicas are timeshifted by a unique random-timeshift vector $\boldsymbol{r}$, where the $j$-th component of the $i$-th random-timeshift $r_j^i \in \boldsymbol{r}^i$ is unique across all random-timeshift vectors $[\boldsymbol{r}^1, \ldots, \boldsymbol{r}^N]$. The randomly-timeshifted state matrix replicas are then horizontally concatenated. As an example, concatenating $N = 2$ randomly-timeshifted state matrix replicas

$$\boldsymbol{S}_{r^1} \frown \boldsymbol{S}_{r^2} = \begin{bmatrix} \boldsymbol{s}_{1-r^1} \\ \ldots \\ \boldsymbol{s}_{K-r^1} \end{bmatrix} \frown \begin{bmatrix} \boldsymbol{s}_{1-r^2} \\ \ldots \\ \boldsymbol{s}_{K-r^2} \end{bmatrix}$$

$$= \begin{bmatrix} s_{1-r_1^1} & \cdots & s_{1-r_n^1, n} \\ \ldots & \ldots & \ldots \\ s_{K-r_1^1} & \cdots & s_{K-r_n^1, n} \end{bmatrix} \frown \begin{bmatrix} s_{1-r_1^2} & \cdots & s_{1-r_n^2, n} \\ \ldots & \ldots & \ldots \\ s_{K-r_1^2} & \cdots & s_{K-r_n^2, n} \end{bmatrix}$$

generates a multi-random-timeshifts state matrix of feature dimension $O = 2n$, illustrated in Fig. 2 at $N = 2$ replicas. Multi-random-timeshifting $N$ replicas mean that states at each node are recalled up to $N$ times, in order to generate the multi-random-timeshifts state matrix $\boldsymbol{S}_{r^1} \frown \ldots \frown \boldsymbol{S}_{r^N}$ of $O = Nn$ features.

We can generate any desired feature dimension $O$ using this method. In the case where the $n$ features of the base state matrix $\boldsymbol{S}$ are not a divisor of the desired feature dimension $O$, then the $N$-th random-timeshifts state matrix replica $\boldsymbol{S}_{r^N}$ is only a partial replica of the first $n - (O \bmod n)$ features.

**Multi-random-timeshifting gives the best performance.** Multi-random-timeshifting results in Table 2 list the best performances when concatenating $N \geq 2$ randomly-timeshifted state matrix $\boldsymbol{S}_{r^N}$ replicas. The optimal max-random-timeshift $R$ is also listed. We report a 62% decrease in error when training on $N = 2$ concatenated randomly-timeshifted state matrices, compared to the single $N = 1$ randomly-timeshifted state matrix $\boldsymbol{S}_{r^1}$. The multi-random-timeshifts state matrix $\boldsymbol{S}_{r^1} \frown \boldsymbol{S}_{r^2}$ as a result of this concatenation has $O = 2n$ features.

Performance up to $N = 6$ replicas shows continued improvement, training on feature dimensions up to $O = 6n$. Additionally, we observe that predictive performance up to $N = 6$ is the same when implementing either multi-uniform-timeshifting or multi-random-timeshifting.

We proceed with multi-random-timeshifting for $N > 6$ replicas. We see that performance continued to improve with increasing replica numbers $N$ beyond what we achieved with multi-uniform-timeshifting. Performance is also seen as the increasingly better overlap of predicted Lorenz $z$-coordinates $\hat{\boldsymbol{y}}$ vs target $\boldsymbol{y}$ (Suppl. Note 1).

We then investigated the parameter space for optimal max-random-timeshift $R$ configurations, illustrated in Fig. 5a. Here, we scanned the max-random-timeshift $R$ against increasing the feature dimension of the multi-random-timeshift state matrix $\boldsymbol{S}_{r_1} \frown \ldots \frown \boldsymbol{S}_{r_N}$. The size of the reservoir is clamped at $n$ nodes, while the number of feature $O$ increases in increments of 1. When focusing on any specific max-random-timeshift $R$ we found that predictive performance generally improves with larger feature dimension $O$, even at partial multiples of $n$.

Another illustration of these data is in Fig. 5b, where we observed that the optimal max-random-timeshift $R$ (maroon dots) generally increases as feature dimension $O$ increases. This could be expected, since a higher number of sampled states from each node must be recalled in order to generate the multi-random-timeshifts state matrix $\boldsymbol{S}_{r_1} \frown \ldots \frown \boldsymbol{S}_{r_N}$. With very high replica numbers up to $N \approx 100$, multi-random-timeshifting continues to improve (teal line). Beyond $N \approx 100$ replicas, we observe the optimal max-random-timeshifts $R$-value consistently at the theoretical minimum $R = N - 1$, and predictive performance no longer improves.

The performance of the clamped reservoir of $n$ nodes at total recall $J = n$ performed just as well as the correspondingly upscaled reservoir of $Nn$ nodes at null recall $J = 0$ indicating that implementing multi-random-timeshifting with total recall $J = n$ is an effective way to increase the feature

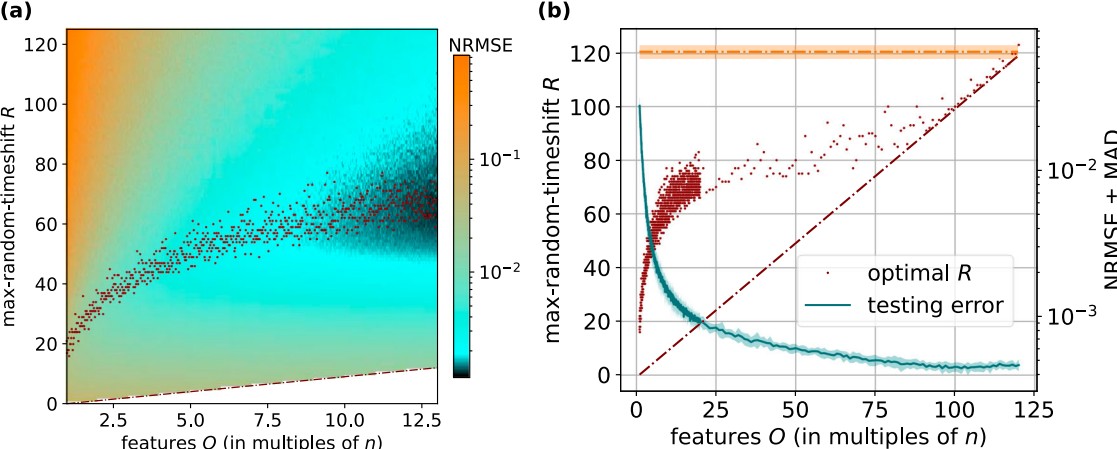

**Fig. 5 | Multi-random-timeshifting on the simulated reservoir computer, Lorenz *x*-to-*z* cross-prediction. a** Scan of feature dimension $O$ (up to $O = 13n$ features) vs max-random-timeshift $R$, for testing error. Maroon dots indicate optimal max-timeshift $R$ at feature dimension $O$. The maroon dashed line indicates the minimum value that the max-random-timeshift $R$ may take at a given feature dimension. **b** Scan of feature dimension $O$ (up to $O = 120n$ features) vs max-random-timeshift $R$, for testing error (teal line). For $O \leq 20n$, data are complete for partial replicas of the state matrix. Maroon dots indicate optimal max-timeshift $R$ at feature dimension $O$. The maroon dashed line indicates the minimum value that the max-random-timeshift $R$ may take at a given feature dimension. The orange dashed line indicates the result of training on the base state matrix $\mathbf{S}$. The shaded area around each line indicates the MAD of the realisation set. NRMSE normalised root mean square error, MAD median absolute deviation.

dimension $O$ and maintain the superior predictive performance of this method, while also clamping the reservoir size to $n$ nodes (Suppl. Note 1).

The simple post-processing and the multi-random-timeshifting methods were established on the Lorenz attractor. These methods were corroborated with the Mackey–Glass $P$-to-$P$ 10-step-ahead forecasting (Suppl. Note 3) and Rössler $x$-to-$z$ cross-prediction (Suppl. Note 4) tasks on the simulated reservoir computer. We consistently observed our post-processing methods improving performance. As expected, increasing the number of replicas $N$ improved performance even further. Moreover, increased recall $J$ resulted in superior performance, while also reducing reservoir node dimension from $Nn$ to $n$.

**Optimising timeshift delays**. Finding the optimal timeshift configurations for our post-processing methods is key to optimal delay embedding. For multi-uniform-timeshifting, we saw that the optimal uniform-timeshifts $\boldsymbol{d}$ vary with increasing $N$. As the number of replicas $N$ increases, each uniform-timeshifts configuration $\{d_2, \dots d_N\} \in \boldsymbol{d}$ must be re-optimised by an $(N-1)$-dimensional scan. To perform this optimisation for all uniform-timeshifts $d$, the number of linear regression operations required exponentially increases by

$$d_{\text{scan}}^{N-1}$$

where $d_{\text{scan}}$ is the maximum of the range of uniform-timeshifts $d \in \boldsymbol{d}$ to be scanned.

For multi-random-timeshifting, only a 1-dimensional scan for the max-random-timeshift $R$ is required even as the number of $N$ replicas increases. The number of linear regression operations required for optimising the max-random-timeshift $R$ increases by

$$R_{\text{scan}} - N + 1$$

where $R_{\text{scan}}$ is the maximum of the range of max-random-timeshifts $R$ to be scanned.

Computational time benchmarks on our simulated reservoir computer are given as an example. We require 190 s to compute a random-mask realisation set, for a post-processed training matrix of $O = 10n$ features from $N = 10$ replicas. Optimising the max-random-timeshift $R$, where the

scanning range of max-random-timeshifts $R_{\text{scan}} = 180$, is calculated as

$$190\,\text{s} * (180 - 10 + 1) = 9\,\text{h}.$$

Comparatively, the computational time required to optimise the multi-uniform-timeshifts $\{d_2, \dots d_{10}\} \in \boldsymbol{d}$, where the scanning range for each uniform-timeshift is $d_{\text{scan}} = 20$, such that $d_{\text{scan}}(N-1) = 180$ matches the scanning max-random-timeshift scanning range $R_{\text{scan}} = 180$, is calculated as

$$190\,\text{seconds} * (20^{10-1}) = 185083714\,\text{years}.$$

Due to this ease of optimisation for larger $Nn$ feature dimensions, multi-random-timeshifting ultimately outperforms multi-uniform-timeshifting in terms of scalability and predictive performance. For similar reasoning, random-timeshifting is preferable to non-random-timeshifting of training features. Finding non-random, optimal-timeshifts $[r_1, \dots, r_n] \in \boldsymbol{r}$ quickly becomes computationally intractable as $n$ increases, due to the $n$-dimensional scan required to optimise every optimal-timeshift $r \in \boldsymbol{r}$ applied to each feature. First-order approximations of optimal time-shifts have been previously implemented[6].

**Translation to a physical reservoir**

Post-processing methods were established, characterised and corroborated on our simulated reservoir computer. Implementing our post-processing methods improves predictive performance, and need only be applied after the readout is generated. Thus, in this section we translate our methods to a physical reservoir implemented in a photonic system, enabling high-speed operation using GHz-rate data inputs. To accomplish this, we took our readout of a previously published VCSEL-based photonic reservoir system[17] that was used for high-dimensional mapping of input. Figure 6a illustrates the experimental setup. When compared to the simulated reservoir the presence of an internal time delay ($\tau$), as well as masking ($m$) and scaling ($g$) of the input is reflected in the experimental setup. The key difference between the simulated reservoir and the experimental reservoir is the number of nodes $n$. Despite these differences in the readout, post-processing methodologies remained unaffected and could be implemented as established.

In addition to the physical reservoir readout, we took our driving $P$-coordinates of the input series, corresponding to a Mackey–Glass attractor. This enabled us to choose our prediction task, which we set as $P$-to-$P$ 10-step-ahead forecasting.

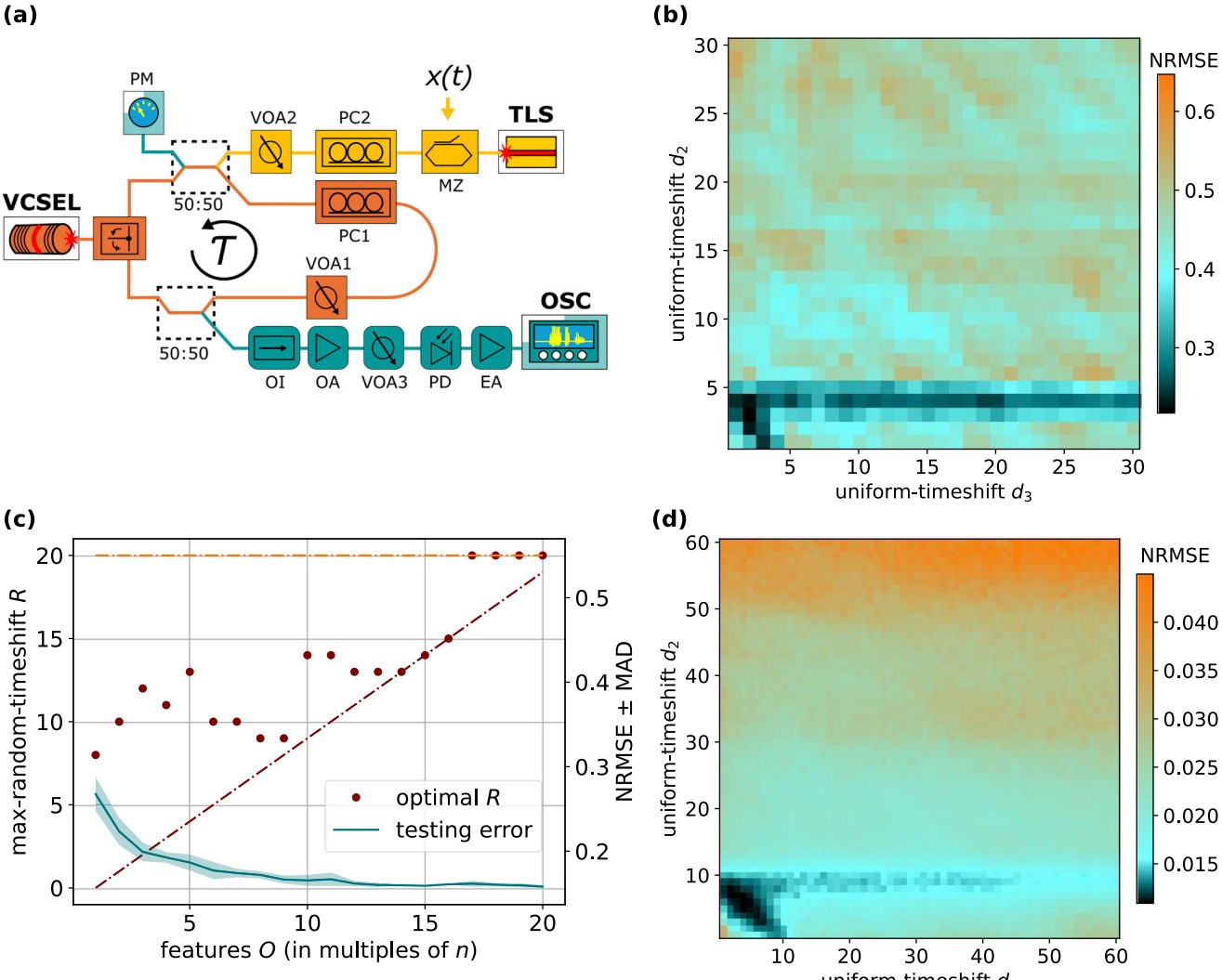

**Fig. 6 | Timeshifting on the physical photonic reservoir computer, Mackey–Glass P-to-P 10-step-ahead forecasting. a** Experimental setup of our physical reservoir[17]. The device receives and processes the input signal $x(t)$ (gold components) before feeding into the reservoir that performs nonlinear mapping (orange components). The reservoir state is observed and collected as the physical reservoir's readout (teal components). **b** Scan of multi-uniform-timeshifts $d_2$ vs $d_3$, for testing error at $N = 3$ replicas and $O = 3n$ features. **c** Scan of feature dimension $O$ (up to $O = 20n$ features) vs max-random-timeshift $R$, for testing error (teal line). Maroon dots indicate

optimal max-timeshift $R$ at feature dimension $O$. The maroon dashed line indicates the minimum value that the max-random-timeshift $R$ may take at a given feature dimension. The orange dashed line indicates the result of training on the base state matrix $S$. The shaded area around each line indicates the MAD of the realisation set. **d** Simulated reservoir comparison to panel (**b**). NRMSE normalised root mean square error, MAD median absolute deviation.

In the "uniform-timeshifting, $S \frown \dots \frown S_{d_N^\Sigma}$" column of Table 3, replica number $N = 1$ corresponds to training on the base state matrix $S$, without post-processing. We report a 45% decrease in error when training on the $N = 2$ uniform-timeshift state matrix $S \frown S_{d_2}$. This improved performance is consistent with improvements seen in the simulated computer. Training on $N = 3$ uniform-timeshifts state matrix $S \frown S_{d_2} \frown S_{d_3^\Sigma}$ gives a further 27% decrease in error from $N = 2$. Training on $N \geq 3$ uniform-timeshifts state matrix $S \frown \dots \frown S_{d_N^\Sigma}$ indicates progressively improving performance. Uniform timeshifts $\{d_1, \dots d_N\}$ must be re-optimised with increasing $N$ replicas, which is also consistent with the simulated reservoir computer. Additionally, Fig. 6b shows the parameter space of optimal configurations (dark teal/black) for uniform timeshifts $d_2$ and $d_3$. The optimal configuration space on the physical reservoir follows a similar configuration space observed on the simulated reservoir computer for the same Mackey–Glass P-to-P 10-step-ahead-forecasting task, shown in Fig. 6d.

In the "random-timeshifting, $S_{r^1} \frown \dots \frown S_{r^N}$" column of Table 3, starting at $N = 1$ replica we report that optimal random-timeshifting leads to a 51%

decrease in error compared to base training. Consistent with observations on the simulated reservoir system, Fig. 6c illustrates how training on $N \geq 2$ multi-random-timeshifts state matrices $S_{r^1} \frown \dots \frown S_{r^N}$ progressively improves predictive performance (teal line). We also observed that the optimal max-random-timeshift $R$ (maroon dots) generally increases with replica number $N$, as was observed on the simulated reservoir computer.

Mackey–Glass P-to-P 1-step-ahead forecasting was also carried out (Suppl. Note 5). Our results for training on the base state matrix $S$ without post-processing corroborate with previously published results for this task[17].

Taken together, these consistencies suggest that post-processing methods established using simulated reservoirs can be translated to experimentally realised readout data generated from a physical reservoir system.

## Conclusion

We established simple post-processing methods on our simulated reservoir computer for a variety of chaotic attractor prediction tasks. These methods introduced delayed versions of reservoir states for training, as inspired by delay embedding.

**Table 3 | Physical photonic reservoir computer results of Mackey–Glass *P*-to-*P* 10-step-ahead forecasting**

| N replicas | O features | Uniform-timeshifting, $S^\frown ... ^\frown S_{d_N^\Sigma}$ | | | | | | Random-timeshifting, $S_{r^1}^\frown ... ^\frown S_{r^N}$ | |
|---|---|---|---|---|---|---|---|---|---|
| | | NRMSE | $d_2$ | $d_3$ | $d_4$ | $d_5$ | $d_6$ | NRMSE ± MAD | R |
| 1 | n | 0.55* | – | – | – | – | – | 0.27 ± 0.02 | 8 |
| 2 | 2n | 0.30 | 4 | – | – | – | – | 0.22 ± 0.02 | 10 |
| 3 | 3n | 0.22 | 3 | 2 | – | – | – | 0.20 ± 0.01 | 12 |
| 4 | 4n | 0.19 | 3 | 1 | 1 | – | – | 0.193 ± 0.006 | 11 |
| 5 | 5n | 0.17 | 3 | 1 | 1 | 2 | – | 0.19 ± 0.01 | 13 |
| 6 | 6n | 0.15 | 3 | 1 | 1 | 7 | 9 | 0.18 ± 0.01 | 10 |
| 7 | 7n | – | – | – | – | – | – | 0.174 ± 0.005 | 10 |
| 10 | 10n | – | – | – | – | – | – | 0.165 ± 0.006 | 14 |
| 20 | 20n | – | – | – | – | – | – | 0.158 ± 0.002 | 20 |

*NRMSE* normalised root mean square error, *MAD* median absolute deviation.

*Result of training on the base state matrix **S**.

Improved performance of training on the optimal uniform-timeshift state matrix, which concatenates a uniformly-timeshifted state matrix to the base state matrix, was partly attributed to its increased feature dimension[35]. Uniform timeshifting can thus be viewed as a computationally inexpensive post-processing alternative to increasing reservoir node dimension for higher-dimensional mapping of the input[36,37], that achieves similar results.

We expanded uniform-timeshifting into multi-uniform-timeshifting, which concatenates uniformly-timeshifted state matrices into a multi-uniform-timeshifts state matrix. The improved performance with increasing state matrix replicas is consistent with delay embedding playing a role. Additionally, the optimal uniform timeshifts are not evenly spaced, indicating that a non-uniform delay embedding is ideal.

Random-timeshifting of base state matrix features was established on our simulated computer. The performance improvement when training on the optimal random-timeshifts state matrix was consistent with previous work[6,7]. We did not observe state matrices with larger covariance rank generally outperform state matrices of smaller covariance rank[7,12,39,40], when scanning max-random-timeshifts at optimal and sub-optimal values. A larger covariance rank is therefore not a complete explanation of why optimal random-timeshifting outperformed training on the base state matrix of equal feature dimension.

Under optimal timeshift delays, we found that the total recall of past states from existing nodes is superior to collecting extra node states from a larger reservoir. This finding inspired multi-random-timeshifting, by randomly recalling past states of a clamped reservoir in order to increase the number of training features. Multi-random-timeshifting performs just as well as multi-uniform-timeshifting when training on equivalent feature dimensions derived from equivalent reservoir sizes. However, given that multi-random-timeshifting is much cheaper to optimise, we were able to reach better performance with this method at feature dimensions that are intractable to optimise for multi-uniform-timeshifting. Multi-random-timeshifting is thus our preferred post-processing method among those investigated in this paper.

We translated our established methods to an experimentally realised physical photonic reservoir system[17], given that our post-processing methods are applied to the training scheme after reservoir states are generated. We showed that our methods improved predictive performance. We believe that integrating multi-random-timeshifting in the standard training scheme of both simulated and physical reservoir computers can improve performance for minimal computational investment.

Our implementation of the multi-random-timeshifting algorithm is applicable to autonomous prediction tasks, such as Lorenz attractor closed-loop predictions[41]. Aside from influencing training performance, the multi-random-timeshifting algorithm introduces delay terms to the trained autonomous system when the reservoir is operated in a closed loop. The influence of these time delays on solutions and the stability of the trained autonomous system remains to be explored in future work.

## Data availability
The simulation data that support the findings of this study are available from the corresponding author upon reasonable request.

## Code availability
Code implementing the timeshifting post-processing algorithms used during the current study is available online[42]. Additional simulation code is available from the corresponding author upon reasonable request.

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

## Acknowledgements

J.J., J.R., A.H., and K.L. acknowledge funding from the European Union's Horizon 2020 programme under grant agreement number 101129904, SPIKEPro. L.J. acknowledges funding from the Carl-Zeiss-Stiftung. A.H. and J.R. acknowledge funding from the UKRI Turing AI Acceleration Fellowships Programme (EP/V025198/1) and support from the Fraunhofer Centre for Applied Photonics, FCAP.

## Author contributions

J.J. performed the simulations. J.R. and A.H. performed the experimental work. J.J. performed the analysis with L.J. and K.L. K.L. supervised this work. All authors contributed to discussing and writing the manuscript.

## Funding

## Competing interests

The authors declare no competing interests.
