## [Transparent Peer Review file · Communications Engineering]

Post-processing methods for delay-embedding and feature scaling of reservoir computers

Corresponding Author: Dr Jonnel Anthony Jaurigue

Version 0:

Reviewer comments:

Reviewer #1

(Remarks to the Author)

Report on

Total recall: Post-processing methods for delay-embedding and feature scaling of reservoir computers

by Jonnel Jaurigue et al

In this work the authors investigate of how concatenating delayed versions of the states of a reservoir computer to its output improves performance. Different approaches are studied, including using all the previous states at a delay d , or randomly chosen states. This idea is not new, as acknowledged in the paper. But the authors carry out an in depth study, which provides new insights. The paper is generally well written. Given that this is -to my knowledge- the first in depth study of this method, and given that it is rather easy to implement, it will help improve the reservoir computing approach, and will have some impact. Overall I find it suitable for publication once the following points are taken into account.

1 The authors acknowledge in a very honest way previous works, in particular concerning “random time shifting”. The “(multi)-uniform timeshifting” method has also I believe been introduced earlier in Ref [25] (E. Picco, P. Antonik, and S. Massar, High speed human action recognition ...) under the name « Time Frames of Interest”.

2 From what I understand, I expect that the multiuniform time shifting should be indifferent to the order of the time shifts. That is $S^{\wedge}S_{d2}^{\wedge}S_{d3}$ should give exactly the same results as $S^{\wedge}S_{d3}^{\wedge}S_{d2}$. However this symmetry is absent in the figures (see eg. Fig 4a, 4b, 8b, 13a, 13b. Either there is a mistake in the figures, or the method is not explained satisfactorily.

3 Figure 13. I don't understand the differences between panels an and b, in particular the vertical axis (and the reference under panel b of a mysterious panel c).

4 Section “Optimising timeshift delays » . I don't understand why the computational cost is $d_SCAN^{\wedge}N-1$. If order does not matter (see above), then it should be the combinatorial factor d_SCAN choose $N-1$. In addition I doubt that scanning all the possible choices is useful. Some form of random sampling should be sufficient, and would drastically decrease the number of years required. Statements like “Moreover, this method is computationally cheap to implement and can outperform all other postprocessing methods. » in the introduction may need revision.

4 I understand why the recall method parametrized by J is only introduced in section III B. However I would recommend mentioning this method at the end of section II B (Methods: Post processing the state matrix). Similarly for “Multi Random Time Shifting” (section II C).

5 I did not understand the difference between multi random time shifting (section II C).and random time shifting. Maybe a more careful explanation and an illustration like in figure 2 would be useful.

6 Figure 2. Is O the total feature dimension, or only the added number of features. There seems to be discrepancy between Fig 2 and the text.

7 The reservoir model should be given explicitly and parameter τ (see figure 17) defined. Also τ is used in the definition of the Mackey-Glass system, so another letter should be used.

A few minor points of English.

8 O is defined below Eq. 1, but first introduced above Eq. 1. Please correct.

9 “Additionally, changing the prediction task from x-to-z cross-prediction to ...” I don’t find “Additionally” the appropriate word. Maybe “Note that ...”.

10 End of same paragraph “... generally outperforms a state matrix of a smaller covariance rank[7, 11, 38, 39] is incomplete. » Consider changing « incomplete”.

11 « In addition to the physical reservoir readout, we were provided the driving P-coordinates of the input series.. » Consider changing “we were provided”, as the experimentalists are coauthors of the paper.

Reviewer #2

(Remarks to the Author)

I co-reviewed this manuscript with one of the reviewers.

Reviewer #3

(Remarks to the Author)

The authors Jaurigue et al. investigated the performance comparison of various post-processing methods on reservoir features. Based on their description, they proposed effective modification strategies to further enhance the performance and computation efficiency. Both simulation results and experimental results were presented to support their claims. Overall, I think this work present useful information to the field of reservoir computing and I have several comments for the authors to be addressed before publication.

1. Section II.A, why there is a vector of ones concatenated after the state matrix S to construct S_{prime} ? Is the state matrix itself not enough for this task?
2. The authors reported results on cross-prediction tasks and prediction tasks on relatively low dimensional datasets. Will the proposed strategies useful for more challenging datasets like spatiotemporal chaotic timeseries and for more challenging scenarios such as autonomous forecasting, where the output of one-time step is taken as the input to the next time step (i.e. close-loop prediction)?
3. The discussion of delay concatenation of reservoir states in this work reminds me a lot of the next generation reservoir computing algorithm [Pyle et al. Phil. Trans. R. Soc. A 379:20200246 (2020); Gauthier et al. Nature Communications 12, 5564 (2021)] and their hardware implementations [Kent et al. Nature Communications 15, 3886 (2024); Cox et al. arXiv:2404.07116v1 (2024); Wang et al. arXiv:2404.07857 (2024); [Wang et al. arXiv:2407.05840v1 (2024)]. I think the authors could comment on the connections and differences to those works, which can make their contributions more appreciable.
4. Can the author provide a clear comparison table about the hyperparameters needed by a standard baseline reservoir computing network and a counterpart with total recall post-processing to the readers?
5. There are some typos about symbols, like the s_k and x_k in superscripts in Section II.A, shouldn't they be in subscripts ?
6. I suggest the authors making their codes and data publicly available to increase the publicity of this work.

Reviewer #4

(Remarks to the Author)

In this manuscript, the authors present a delay embedding post-processing technique to improve the Reservoir Computer (RC) performance. They propose an interesting selection architecture for the feature vector whose components are composed by the current states of some or all nodes in the network plus past states of the same or different nodes. The ratio between the usage of same and different nodes to compose the delayed states is measured by a recall parameter β . The authors explore random time-shifts to explore more time-delays with less computational resources and come up with what they call “multi-random-time-shifting” method, which essentially increase the number of random features selected from past reservoir states. Finally, the authors apply their algorithms to reservoir data obtained from a physical system. The manuscript is well written, well presented and does not present scientific errors. I believe that the current version of this manuscript presents important advances in reservoir computing and is very interesting for the machine learning and nonlinear dynamics communities. I recommend this manuscript to be published on Communications Engineering after the authors address my

comments below.

1. I start with my biggest concern, which is about the application of this method to autonomous prediction. In many parts of the manuscript the authors refer to their algorithm as a post-processing method, which is true for the inference task (Lorenz x -to- z cross-prediction). In this case, the x variable of the system is used as input to drive the reservoir for a long time and the z variable can be inferred afterwards. I believe that for autonomous prediction tasks, the implementation of this method will not be a postprocessing method. Instead, not only the reservoir needs to evolve in each time step but all the calculation involving the past reservoir states and the weights, necessary for the prediction, need to be made before the next time step takes place. Although the authors perform some prediction tasks displayed in the Appendix, I believe that these predictions are not autonomous. The authors should clarify how their predictions are made. To summarize my comment, applying the proposed technique to an autonomous prediction task would introduce latencies to the algorithm that need to be taken in consideration when evaluating the efficiency of the proposed method. Finally, if the authors performed an autonomous prediction (which is not clear if they did or not), a computational cost and performance comparison between the method implemented in this work for prediction and other methods would be beneficial for this manuscript.

2. Similar machine learning approaches that use time delayed variables has been able to learn the target system very efficiently and to identify the system vector field components using a Nonlinear-Vector-Auto-Regression-like approach without the need of an Artificial Neural Network (ANN) (Nat Commun 12, 5564 (2021)) or sparse optimization (Chaos 31, 082101 (2021)), for example. This identification means that some of the learned weights are very small and can be removed using various feature selection methods without hurting the testing error. Examples of such feature selection methods include Lasso regression, support vector machines with kernel trick or simply starting with a lower number of features and adding terms one-by-one, keeping only those terms that reduce substantially the testing error, like proposed by Billings (Billings, S. A. Nonlinear System Identification (John Wiley & Sons, Ltd., 2013)). I believe the manuscript would be strengthen if the authors could provide a comparison between their method and a Lasso regression, for example, which can be performed directly on the features composed by the current and past reservoir states.

3. In the same line as the comment above, one of the main claims of this work is the low computational cost and high performance. It would be interesting if the authors addressed a direct comparison with pros and cons between the authors proposed post-processing delay embedding technique and delay embedding techniques performed direct on data, like for example in the work in this reference: Nat Commun 12, 5564 (2021).

4. In Table III, I notice that, regardless the number of replicas (N), the maximum time delay is always around 21 for the uniform-time-shifting case. It makes me think that important information about the current state of the learning system is somewhat present in 21 steps in the past in the reservoir. As the authors highlight in multiple parts of the text, everything boils down to delay embedding. What is the maximum value allowed by the authors for d_i ? Also, have the authors played with the reservoir hyper-parameters and RC time resolution (integration step or sampling rate for experimental approach) to optimize the network timescale and memory to match the characteristic time of the target system? If so, that would be great to make some comment or show some results in this regard.

5. Regarding results shown in Figs. 10 and 12. When increasing the feature vector dimension O , there should be a moment where the multiple random time-shift should start to decrease its performance because the feature vector would become similar to the one at the multiple uniform time-shift. Is the performed scan (up to $O=125n$) far from this upper limit? Could the authors address this point?

6. What changes between different realizations in the random time-shifting cases? Do the authors only change the RC initial condition and keep the same time-shift/nodes for each feature (initially randomly chosen) or the time-shift/nodes are randomly chosen every new realization?

Version 1:

Reviewer comments:

Reviewer #1

(Remarks to the Author)

I believe the authors have adequately taken into account the reviewer comments.

Reviewer #2

(Remarks to the Author)

The authors have addressed all my comments.

Reviewer #3

(Remarks to the Author)

The authors have addressed my comments in a satisfactory manner.

Reviewer #4

(Remarks to the Author)

The authors have addressed most of my comments from my previous review and clarified that some of the topics I raised will be covered in future work. However, I still believe that comparing the proposed method to other machine learning methods would strengthen this study. Nonetheless, the current version of the manuscript presents important advances in reservoir computing and is relevant to the machine learning and nonlinear dynamics communities, as I mentioned previously. I recommend the new version of the manuscript for publication in Communications Engineering.

Total recall: Post-processing methods for delay-embedding and feature scaling of reservoir computers (Authors' response)

Jonnell Jaurigue^{1,*}, Joshua Robertson², Antonio Hurtado², Lina Jaurigue¹, and Kathy Lüdge¹

¹*Technische Universität Ilmenau, Institut für Physik,
Weimarer Straße 25, 98693 Ilmenau, Germany and*

²*University of Strathclyde, Institute of Photonics,
SUPA Department of Physics, 99 George Street, G1 1RD, Glasgow, UK*

I. REVIEWER 1 AND 2

1.1

1 The authors acknowledge in a very honest way previ-
ous works, in particular concerning “random time shift-
ing”. The “(multi)-uniform timeshifting” method has
also I believe been introduced earlier in Ref [25] (E. Picco,
P. Antonik, and S. Massar, High speed human

Thank you for drawing attention to this oversight. I
have added text at in “I. Introduction” and “IIB. Multi-
uniform-timeshifting” citing “Timesteps Of Interest”.

1.2

2 From what I understand, I expect that the multi-
uniform time shifting should be indifferent to the order of
the time shifts. That is $S^{\wedge}S_{d_2}^{\wedge}S_{d_3}$ should give exactly
the same results as $S^{\wedge}S_{d_3}^{\wedge}S_{d_2}$. However this symmetry
is absent in the figures (see eg. Fig 4a, 4b, 8b, 13a, 13b).
Either there is a mistake in the figures, or the method is
not explained satisfactorily.

We can see how our notation can be misleading and
have changed the nomenclature accordingly in “IIB.
Multi-uniform-timeshifting”. An example concatenated
state matrix of delayed states S_{d_4} is now $S_{d_4^{\Sigma}}$ to better
emphasise that this timeshifted matrix is delayed by a
sum of iterative timesteps into the past $d_2 + d_3 + d_4$.

We have also added a clarifying example with numbers
regarding how this implementation of multi-timeshifting
works. With the clarifying example, it should now be
clearer that swapping the values of stepsize d_2 and d_3
do not yield the same pair of timeshifted state matrix
replicas $S_{d_2^{\Sigma}}$ and $S_{d_3^{\Sigma}}$.

For completeness, the horizontal concatenation order
of the state matrices indeed does not matter; that is,
 $S^{\wedge}S_{d_2^{\Sigma}}^{\wedge}S_{d_3^{\Sigma}}$ and $S^{\wedge}S_{d_3^{\Sigma}}^{\wedge}S_{d_2^{\Sigma}}$ give the same results,
provided $S_{d_2^{\Sigma}}$ and $S_{d_3^{\Sigma}}$ are the same pair of timeshifted
state matrix replicas in each example, defined by step-
size d_2 and d_3 (which cannot be swapped indifferently).

* Email address: jonnell-anthony.jaurigue@tu-ilmenau.de

1.3

3: Figure 13. I don't understand the differences be-
tween panels an and b, in particular the vertical axis
(and the reference under panel b of a mysterious panel
c).

Thank you for pointing out this mistake; Text in Figure
13 is changed from referencing panel (c) to panel (a). The
comparison between the hardware-implemented reservoir
and the simulated reservoir should now make sense.

1.4

4 Section “Optimising timeshift delays” I don't under-
stand why the computational cost is $d_{\text{SCAN}}^{\wedge}N-1$. If
order does not matter (see above), then it should be the
combinatorial factor d_{SCAN} choose $N-1$. In addition
I doubt that scanning all the possible choices is useful.
Some form of random sampling should be sufficient, and
would drastically decrease the number of years required.
Statements like “Moreover, this method is computationally
cheap to implement and can outperform all other
postprocessing methods.” in the introduction may need
revision.

Regarding the exponential scanning equation:

For our implementation of multi-uniform-timeshifting,
we were inspired by delay-embedding. We start with a
fixed timeshift d , and take $(N - 1)$ iterative timeshifts d
into the past, so that the earliest state matrix replica is
 $(N - 1)d$ steps into the past.

From here, we also allowed the size of timeshift d to
vary, for $N = 10$ iterative timeshifts into the past. A
 d_{scan} value of 20 (as in the example) means that each
timeshift d can be any size from 1-20 input-steps. The
earliest replica $S_{d_N^{\Sigma}}$ is timeshifted $d_2 + d_3 + \dots + d_N$ input-
steps into the past, which can be anywhere between $(N - 1) = 9$
and $20(N - 1) = 180$ input-steps. This also means that any
concatenated timeshift is at maximum $d_{\text{scan}} = 20$ input-steps
away from its nearest neighbour. Scanning all possible iterative
timeshift configurations is thus d_{scan}^{N-1} .

A combinatorial equation does not fit, based on how
 d_{scan} is defined. If we were to define d_{scan} in this equa-
tion as the timespan in which we collect past states (i.e.
the timespan up to 180 input-steps into the past) then
we do not capture the limitation that any concatenated

timeshift is at maximum $d_{\text{scan}} = 20$ input-steps away from its nearest neighbour.

Regarding the requirement to scan d configurations for optimisation, we state the following in the text:

“We do not observe multiple or repeating optimal zone patterns, illustrating that configurations must be specifically chosen for superior performance. This indicates the importance of thorough scans to determine the optimal uniform-timeshifts $d \in \mathbf{d}$ configurations.”

Regarding random sampling possibilities:

Random sampling methods were what lead us to the (multi)-random-timeshifting methods. I have added text at “IIIA. Random-timeshifting improved performance” to better illustrate this workflow/narrative.

1.4b

I understand why the recall method parametrized by J is only introduced in section III B. However I would recommend mentioning this method at the end of section II B (Methods: Post processing the state matrix). Similarly for “Multi Random Time Shifting” (section II C).

Indeed, earlier drafts had mentioned recall and multi-random-timeshifting in the methods. Having considered both options, we opted for the current version as is printed. That is, we introduce recall and multi-random timeshifting after presenting the methods and results of the “groundwork” post-processing methodologies.

1.5

I did not understand the difference between multi random time shifting (section II C).and random time shifting. Maybe a more careful explanation and an illustration like in figure 2 would be useful.

We have expanded Figure 2 to more clearly differentiate the post-processing methodologies that are presented.

1.6

Figure 2. Is O the total feature dimension, or only the added number of features. There seems to be discrepancy between Fig 2 and the text.

The definition of O is consistent in the preprint text:

“The number of columns of state matrix \mathbf{S} is the feature dimension O .”

“For base training without further post-processing, the feature dimension O equals the node dimension n ”

“In this way, the uniform-timeshift state matrix $\mathbf{S} \hat{\sim} \mathbf{S}_{d_2}$ that is used for training and testing has an increased feature dimension of $O = 2n$.”

1.7

The reservoir model should be given explicitly and parameter tau (see figure 17) defined. Also tau is used in the definition of the Mackey-Glass system, so another letter should be used.

Thank you for the suggestions. The Mackey-Glass parameter is changed from τ to τ_{MG} . We have also rearranged the end of the Methods section, adding the header “IID. Reservoir systems” to better emphasise the reservoir systems we used.

1.8

A few minor points of English. O is defined below Eq. 1, but first introduced above Eq. 1. Please correct.

“Section IIA. Reservoir computer model” has been revised for better readability and consistency.

1.9

“Additionally, changing the prediction task from x-to-z cross-prediction to ...” I don’t find “Additionally” the appropriate word. Maybe “Note that ...”.

Thank you. I have changed the text to “Note that”, as suggested.

1.10

End of same paragraph “... generally outperforms a state matrix of a smaller covariance rank[7, 11, 38, 39] is incomplete. Consider changing “incomplete”.

“is incomplete” changed to “warrants further investigation”.

1.11

“In addition to the physical reservoir readout, we were provided the driving P-coordinates of the input series.. ” Consider changing “we were provided”, as the experimentalists are coauthors of the paper.

Thank you for pointing out this mistake. Text has been edited in “IID. Translation to a physical reservoir” to correct this.

II. REVIEWER 2

Reviewer 2 (Remarks to the Author):

I co-reviewed this manuscript with one of the reviewers.

III. REVIEWER 3

Reviewer #3 (Remarks to the Author):

The authors Jaurigue et al. investigated the perfor-²²⁵
 175 mance comparison of various post-processing methods on
 reservoir features. Based on their description, they pro-
 posed effective modification strategies to further enhance
 the performance and computation efficiency. Both simu-
 180 lation results and experimental results were presented to²³⁰
 support their claims. Overall, I think this work present
 useful information to the field of reservoir computing and
 I have several comments for the authors to be addressed
 before publication.

3.1

235

185 1. Section II.A, why there is a vector of ones concate-
 nated after the state matrix S to construct S_{prime} ? Is
 the state matrix itself not enough for this task?

“Section IIA. Reservoir computer model” has been re-²⁴⁰
 190 vised for better readability and consistency, including
 how we describe the vector of ones. The vector of ones is
 the bias, which is used for the linear regression algorithm.

3.2

245

2. The authors reported results on cross-prediction
 tasks and prediction tasks on relatively low dimensional
 195 datasets. Will the proposed strategies useful for more
 challenging datasets like spatiotemporal chaotic time-
 series and for more challenging scenarios such as au-
 tonomous forecasting, where the output of one-time step
 is taken as the input to the next time step (i.e. close-loop
 200 prediction)?

The methods we propose are primarily of relevance
 for tasks that require the reservoir computing algorithm
 to perform a delay embedding. Whether or not this is²⁵⁵
 205 the case often depends on how a particular task is con-
 structed. For example, a common task is autonomous
 forecasting of the Lorenz 63 system. If all three vari-
 ables are provided as input then the state of the Lorenz
 system is fully defined at each point in time, and added
 output delay will not have a significant effect (slight im-²⁶⁰
 210 provement can be expected with small delays due to the
 increase in the effective feature dimension). However, if
 the reservoir is only being provided with one input vari-
 able, then delay embedding becomes important and the
 methods we have presented are relevant. The challenge²⁶⁵
 215 in autonomous forecasting lies in the lack of predictabil-
 ity whether the desired solutions will be stable in the
 autonomous system. In the context of the methods pre-
 sented here, the question would then be if the resulting
 delay terms in the trained autonomous systems influence²⁷⁰
 220 the stable of the desired solutions, i.e. whether the out-
 put delays result in more/less spurious Lyapunov expo-

nents. This is outside the scope of this work and warrants
 a study in itself.

As to spatiotemporal chaotic systems, in principle it
 is again a question of what input information is be-
 ing provided. If delay embedding is needed, especially
 on timescales longer than the memory of the reservoir,
 then we see no reason why the methods described in the
 manuscript should not be applicable.

Autonomous prediction was not explored in this pa-
 per. However, post-processing methods for autonomous
 prediction is the subject of future work. We have added
 additional text at the end of “IV. CONCLUSION” to
 address this.

3.3

3. The discussion of delay concatenation of reservoir
 states in this work reminds me a lot of the next gener-
 ation reservoir computing algorithm [Pyle et al. Phil.
 Trans. R. Soc. A 379:20200246 (2020); Gauthier et al.
 Nature Communications 12, 5564 (2021)] and their hard-
 ware implementations [Kent et al. Nature Communica-
 tions 15, 3886 (2024); Cox et al. arXiv:2404.07116v1
 (2024); Wang et al. arXiv:2404.07857 (2024); [Wang
 et al. arXiv:2407.05840v1 (2024)]. I think the authors
 could comment on the connections and differences to
 those works, which can make their contributions more
 appreciable.

Next generation reservoir computing is nonlinear vec-
 tor regression. The fundamental difference is the man-
 ner in which nonlinear transforms of the input sequences
 are constructed. In reservoir computing, the nonlinear
 transforms arise due to the reservoir and the random in-
 put weights, whereas in nonlinear vector regression the
 features are chosen explicitly.

3.4

4. Can the author provide a clear comparison table
 about the hyperparameters needed by a standard base-
 line reservoir computing network and a counterpart with
 total recall post-processing to the readers?

With respect to recall J , we do note the change in the
 number of reservoir nodes n , depending on whether the
 reservoir is rescaled with respect to recall J . However,
 other hyperparameters are the same.

For the rest of the paper, the reservoir computing net-
 work used for all simulations is the same, with or without
 post-processing. The post-processing methods are to do
 with how the state matrix is rearranged or augmented
 before the linear regression training step.

Hyperparameters of the reservoir computer simulation
 using the single optical amplifier model are provided in
 “TABLE II. Reservoir system parameters” and are valid
 for all simulations.

3.5

5. There are some typos about symbols, like the s_k and x_k in superscripts in Section II.A, shouldn't they be in subscripts ?

“Section IIA. Reservoir computer model” has been revised for better readability and consistency.

3.6

6. I suggest the authors making their codes and data publicly available to increase the publicity of this work.

Our code for the simulated reservoir computer and post-processing algorithm will be made publicly available.

IV. REVIEWER 4

Reviewer #4 (Remarks to the Author):

In this manuscript, the authors present a delay embedding post-processing technique to improve the Reservoir Computer (RC) performance. They propose an interesting selection architecture for the feature vector whose components are composed by the current states of some or all nodes in the network plus past states of the same or different nodes. The ratio between the usage of same and different nodes to compose the delayed states is measured by a recall parameter J . The authors explore random time-shifts to explore more time-delays with less computational resources and come up with what they call “multi-random-time-shifting” method, which essentially increase the number of random features selected from past reservoir states. Finally, the authors apply their algorithms to reservoir data obtained from a physical system. The manuscript is well written, well presented and does not present scientific errors. I believe that the current version of this manuscript presents important advances in reservoir computing and is very interesting for the machine learning and nonlinear dynamics communities. I recommend this manuscript to be published on Communications Engineering after the authors address my comments below.

4.1

1. I start with my biggest concern, which is about the application of this method to autonomous prediction. In many parts of the manuscript the authors refer to their algorithm as a post-processing method, which is true for the inference task (Lorenz x -to- z cross-prediction). In this case, the x variable of the system is used as input to drive the reservoir for a long time and the z variable can be inferred afterwards. I believe that for autonomous prediction tasks, the implementation of this method will not be a postprocessing method. Instead,

not only the reservoir needs to evolve in each time step but all the calculation involving the past reservoir states and the weights, necessary for the prediction, need to be made before the next time step takes place. Although the authors perform some prediction tasks displayed in the Appendix, I believe that these predictions are not autonomous. The authors should clarify how their predictions are made. To summarize my comment, applying the proposed technique to an autonomous prediction task would introduce latencies to the algorithm that need to be taken in consideration when evaluating the efficiency of the proposed method. Finally, if the authors performed an autonomous prediction (which is not clear if they did or not), a computational cost and performance comparison between the method implemented in this work for prediction and other methods would be beneficial for this manuscript.

The application for autonomous prediction tasks, such as using Lorenz x -coordinates input-steps to generate the next Lorenz- $x+1$ -coordinate, is the same as for inference. A training input of Lorenz x -coordinates will drive the reservoir for a long time and one trains, on an open loop, the weights that predict the Lorenz $x+1$ -coordinate given the current Lorenz x -coordinate. Delayed states can be concatenated according to the post-processing method used.

When the system is closed to run autonomously, the Lorenz $x+1$ -coordinate to be used as input will be predicted by the model using the current and concatenated past reservoir states $(s_{i,1}, \dots, s_{i,N_n}) \in \mathbf{s}_i$, to generate the next input step \hat{x}_{i+1} according to $\mathbf{s}_i \mathbf{w} = \hat{x}_{i+1}$.

Essentially, the closed loop reservoir computer using concatenated delayed states will be analogous (but not identical) to using a reservoir with internal delay.

Concatenating the reservoir states will not introduce latency according to how we implemented the algorithm. Let us use an example multi-uniform-timeshifts configuration of $N = 3$ uniform timeshifts $d_2 = 2$ and $d_3 = 8$. At any i -th input-step we observe the current vector of reservoir states \mathbf{s}_i driven by the input-step x_i . This vector of reservoir states \mathbf{s}_i is recorded into the final state matrix at row i , row $i+d_2$ and row $i+d_2+d_3$. This means that for any future j -th input-step, the next input step can be predicted autonomously using the currently observed state vector \mathbf{s}_j , as well as the \mathbf{s}_{j-d_2} and $\mathbf{s}_{j-d_2-d_3}$ state vectors previously concatenated to the j -th row of the final state matrix at input-step $j-d_2$ and input-step $j-d_2-d_3$.

Including delays in the output via the methods proposed in our manuscript will result in delay terms in the resulting system when run autonomously. Just like with any system of delayed equations, this will mean that states up to the longest delay time need to be stored, other than that there is no difference in the implementation compared to the common autonomous reservoir computing implementations. However, the delay terms can influence the stability of solutions in the trained autonomous system. As mentioned in response to one of the

380 questions of reviewer 3, in the context of the methods pre-
 385 sented here, the question would be if the resulting delay
 390 terms in the trained autonomous systems influence the
 395 stable of the desired solutions, i.e. whether the output
 400 delays result in more/less spurious Lyapunov exponents.
 405 This is outside the scope of this work and warrants a
 410 study in itself.

415 However, we wish to also make our stance clear, that
 420 we do not agree that the applicability of our methods to
 425 autonomous predictions should be concerning at all. Au-
 430 tonomous predictions are one way of operating a trained
 435 prediction algorithm, which may or may not be suitable
 440 depending on the task. For real world tasks often data
 445 is continuously measured in which case open-loop predic-
 450 tions are more suitable, e.g. in predictive maintenance.
 455 If long prediction horizons are the goal for a particular
 460 task, then often direct multi-step-ahead predictions can
 465 lead to more accurate results. And, for cross-prediction
 470 tasks autonomous predictions are not suitable.

475 Post-processing methods for autonomous prediction is
 480 the subject of future work. We have added additional
 485 text at the end of “IV. CONCLUSION” to emphasise
 490 this.

4.2

495 2. Similar machine learning approaches that use time
 500 delayed variables has been able to learn the target sys-
 505 tem very efficiently and to identify the system vector field
 510 components using a Nonlinear-Vector-Auto-Regression-
 515 like approach without the need of an Artificial Neu-
 520 ral Network (ANN) (Nat Commun 12, 5564 (2021)) or
 525 sparse optimization (Chaos 31, 082101 (2021)), for exam-
 530 ple. This identification means that some of the learned
 535 weights are very small and can be removed using various
 540 feature selection methods without hurting the testing er-
 545 ror. Examples of such feature selection methods include
 550 Lasso regression, support vector machines with kernel
 555 trick or simply starting with a lower number of features
 560 and adding terms one-by-one, keeping only those terms
 565 that reduce substantially the testing error, like proposed
 570 by Billings (Billings, S. A. Nonlinear System Identifica-
 575 tion (John Wiley & Sons, Ltd., 2013)). I believe the
 580 manuscript would be strengthen if the authors could pro-
 585 vide a comparison between their method and a Lasso re-
 590 gression, for example, which can be performed directly on
 595 the features composed by the current and past reservoir
 600 states.

605 Thank you for the comment. The reviewer’s sugges-
 610 tion of incorporating further optimisation methods (such
 615 as Lasso regression/L1-regularisation) and compare their
 620 effects is in the scope of future manuscripts, for the fol-
 625 lowing reasons:

630 We wanted to compare the effect of delayed-states con-
 635 catenation post-processing methods on predictive perfor-
 640 mance. We kept the reservoir computer architecture con-
 645 sistent for this comparison, such as the training algorithm

using ordinary least squares (with L2-regularisation λ) in
 matrix form. For the same reason, reservoir hyperparam-
 eters defining the single optical amplifier were also kept
 consistent, in order to isolate the effect of the different
 post-processing methods.

The conclusion of this manuscript is that multi-
 random-timeshifting is the optimal post-processing
 method. Future work that focuses only on multi-random-
 timeshifting would then explore its implementation on
 different reservoir computer architectures. This can in-
 clude the reviewer’s suggested training and feature selec-
 tion algorithms. For the same reason, we can explore dif-
 ferent reservoir hyperparameters, such as internal reser-
 voir delay, and see their effect. For example, we can
 remove internal reservoir delay completely, to see how
 delayed concatenation affects/compensates for a lack of
 internal reservoir delay.

4.3

3. In the same line as the comment above, one of the
 main claims of this work is the low computational cost
 and high performance. It would be interesting if the au-
 thors addressed a direct comparison with pros and cons
 between the authors proposed post-processing delay em-
 bedding technique and delay embedding techniques per-
 formed direct on data, like for example in the work in
 this reference: Nat Commun 12, 5564 (2021).

Thank you for the insightful suggestion; indeed as reas-
 oned above this comparison would fit well with future
 work. The conclusion of this manuscript is that multi-
 random-timeshifting is the optimal delay-embedding-
 inspired post-processing method, in the context of state
 matrix concatenation approaches on fixed reservoir com-
 puter architectures. Future work that explores multi-
 random-timeshifting in the context of different reservoir
 computer architectures will also have the scope to address
 the reviewer’s suggestion of comparing multi-random-
 timeshifting to different machine learning methods.

4.4

4. In Table III, I notice that, regardless the number of
 replicas (N), the maximum time delay is always around
 21 for the uniform-time-shifting case. It makes me think
 that important information about the current state of
 the learning system is somewhat present in 21 steps in
 the past in the reservoir. As the authors highlight in
 multiple parts of the text, everything boils down to de-
 lay embedding. What is the maximum value allowed by
 the authors for d_i ? Also, have the authors played with
 the reservoir hyper-parameters and RC time resolution
 (integration step or sampling rate for experimental ap-
 proach) to optimize the network timescale and memory
 to match the characteristic time of the target system? If
 so, that would be great to make some comment or show

some results in this regard. We can see how our notation can be misleading and have changed the nomenclature accordingly in “IIB. Multi-uniform-timeshifting”.

The repeating timeshift of 21 seen in “TABLE III”⁵³⁰ does not indicate the maximum time delay of 21 input-steps into the past from the current input step. This is because, for example, d_3 indicates the number of input-steps into the past from d_2 . So for $N = 3$, when $d_2 = 13$ and $d_3 = 21$ then the maximum timeshift delay is⁵³⁵ $d_2 + d_3 = 13 + 21 = 34$ input-steps from the current step; this also means that the 21 input-steps in to the past is not the optimal timeshift for $N = 3$ as it was for $N = 2$.

The maximum value we allowed for d_i during (multi)-uniform-timeshifting scans d_{scan} is 60 for d_2 , d_3 and d_4 , for scans up to $N = 4$ replicas. Thus the range of input-steps that can be concatenated is 240 input-steps into the past at $N = 4$ replicas. The parameter scan space is larger than what is displayed on our graphs as we wanted to highlight the optimal configuration zones that we found.⁵⁴⁵

As addressed in 4.2 and 4.3, the reservoir computer architecture remained fixed in order to compare different post-processing methods. The scope of future work will explore multi-random-timeshifting in the context of different reservoir architectures. This includes the effect of internal delay (or no delay), and different tasks including autonomous closed loop systems. We have added additional text at the end of “IV. CONCLUSION” to emphasise this.⁵⁵⁵

4.5

5. Regarding results shown in Figs. 10 and 12.⁵⁶⁰ When increasing the feature vector dimension O , there should be a moment where the multiple random time-shift should start to decrease its performance because the feature vector would become similar to the one at the multiple uniform time-shift. Is the performed scan⁵⁶⁵ (up to $O = 125n$) far from this upper limit? Could the authors address this point?

The maroon dashed-line in Figures 10 and 12 indicate the minimum value that the max-random-timeshift R may take at a given feature dimension. This is indeed

equivalent to a multi-uniform-timeshifting model, where all iterative uniform-timeshift-stepsizes d_2, \dots, d_N are set to 1.

We also addressed the upper limits of multi-random-timeshifting on the performed scan of Figure 10 (up to $O = 125n$) with the text: “With very high replica numbers up to $N \approx 100$, multi-random-timeshifting continues to improve (teal line). Beyond $N \approx 100$ replicas, we observe the optimal max-random-timeshifts R value consistently at the theoretical minimum $R = N - 1$, and predictive performance no longer improves.”

4.6

6. What changes between different realizations in the random time-shifting cases? Do the authors only change the RC initial condition and keep the same time-shift/nodes for each feature (initially randomly chosen) or the time-shift/nodes are randomly chosen every new realization?

For each new realisation of the simulated reservoir we wrote:

“Given that we apply a random mask on the driving input for our simulated reservoir[1], each reservoir computer configuration undergoes repeat realisations, with each realisation using a different random mask. When applicable, each realisation also uses a different vector of random-timeshifts \mathbf{r} ”

For each new realisation of the physical photonic reservoir we wrote:

“Multiple realisations were required in post-processing when using different vectors of random-timeshifts \mathbf{r} ” on the single hardware-realised readout.

Reservoir computer initial conditions, input and target series, and hyperparameter architectures are consistent between realisations.

We are currently exploring the effect of random weights versus optimised weights for autonomous systems[2] and future work will explore the interaction between random/optimised weights and multi-random-timeshifts. We have added additional text at the end of “IV. CONCLUSION” to emphasise this.

V. REFERENCES

-
- [1] L. Jaurigue and K. Lüdge, Reducing reservoir computer hyperparameter dependence by external timescale tailoring, *Neuromorph. Comput. Eng.* **4**, 014001 (2024).
 [2] L. Jaurigue, Chaotic attractor reconstruction using small

reservoirs—the influence of topology, *Mach. Learn.: Sci. Technol.* **5**, 035058 (2024).